# Understanding pectin cross-linking in plant cell walls
Irabonosi Obomighie [1], Iain J. Prentice[2], Peter Lewin-Jones [3], Fabienne Bachtiger[2], Nathan Ramsay[1], Chieko Kishi-Itakura[1], Martin W. Goldberg[1], Tim J. Hawkins[1], James E. Sprittles [3], Heather Knight [1] ✉ & Gabriele C. Sosso [2] ✉

Pectin is a major component of plant cells walls. The extent to which pectin chains crosslink with one another determines crucial properties including cell wall strength, porosity, and the ability of small, biologically significant molecules to access the cell. Despite its importance, significant gaps remain in our comprehension, at the molecular level, of how pectin cross-links influence the mechanical and physical properties of cell walls. This study employs a multidisciplinary approach, combining molecular dynamics simulations, experimental investigations, and mathematical modelling, to elucidate the mechanism of pectin cross-linking and its effect on cell wall porosity. The computational aspects of this work challenge the prevailing egg-box model, favoring instead a zipper model for pectin cross-linking, whilst our experimental work highlights the significant impact of pectin cross-linking on cell wall porosity. This work advances our fundamental understanding of the biochemistry underpinning the structure and function of the plant cell wall. This knowledge has important implications for agricultural biotechnology, informing us about the chemical properties of plant pectins that are best suited for improving crop resilience and amenability to biofuel extraction by modifying the cell wall.

Unlike animal cells, plant cells are surrounded by a cell wall, which provides strength and protection, whilst allowing growth[1]. As well as its structural role, the cell wall constitutes an important barrier that determines what can and cannot gain access to the cell. It is porous, allowing the passage of water and gases but the degree of porosity may vary, and this is what determines which small molecules are admitted to the cell or impeded from entry. The primary cell wall of plants is composed of complex carbohydrates including pectins, cellulose and hemicelluloses and embedded proteins with structural and enzymatic functions[2]. Pectins are characterised by a galacturonic acid (GalA)-rich backbone and form covalent links with each other[3], resulting in the formation of the complex 3-D structure of the cell wall that governs many properties, including mechanical strength and wall porosity[3–6]. The first estimates of cell wall porosity demonstrated that the wall provided a more significant barrier to the passage of small molecules than was previously assumed, suggesting that control of porosity could regulate cellular communication[7]. Nowadays, understanding and predicting plant cell wall pore size is critical to plant biotechnology and agriculture. Small cell wall pore sizes can restrict the entry and efficacy of cell wall-degrading enzymes, such as cellulases, used by invasive pathogens[8], which cause phenomenal crop losses worldwide[9]. Conversely, for the same reason, porous walls are more amenable to biofuel production by the process of saccharification[10,11].

Pectin gelation is affected by cross-links forming between pectin chains or their sugar side chains. Homogalacturonan (HG) pectins, which represent the majority of cell-wall pectin, primarily cross-link to one another via calcium ions ($Ca^{2+}$) between demethylesterified (DM) carboxyl groups, as illustrated in Fig. 1. This cross-linking mechanism is thought to lead to the aggregation of HG chains via the so-called "egg-box" model[12,13], although recent work[14,15] has put the validity of this model into question. The degree of pectin DM is determined by the action of a large family of enzymes, the pectin methylesterases (PMEs)[16,17] and by numerous PME inhibitors (PMEIs)[18]. Demethylesterfied HG can also link to the other pectic domains and xyloglucans[3]. The more complex pectic domain rhamnogalacturonan II (RGII) chains cross-link to one another via borate diester linkages between adjacent sugar side chains[19]. In each case, the degree of cross-linking affects porosity, i.e., the nanometre to micrometre scale spaces within the wall[20].

In this work, we have leveraged a multidisciplinary approach to shed light onto the mechanism of cross-linking of pectin and to establish a connection between the structure of pectin and the porosity of the cell wall.

[1]Department of Biosciences and Durham Centre for Crop Improvement Technology, Durham University, Durham, UK. [2]Department of Chemistry, University of Warwick, Coventry, UK. [3]Warwick Mathematics Institute, University of Warwick, Coventry, UK. ✉e-mail: p.h.knight@durham.ac.uk; g.sosso@warwick.ac.uk

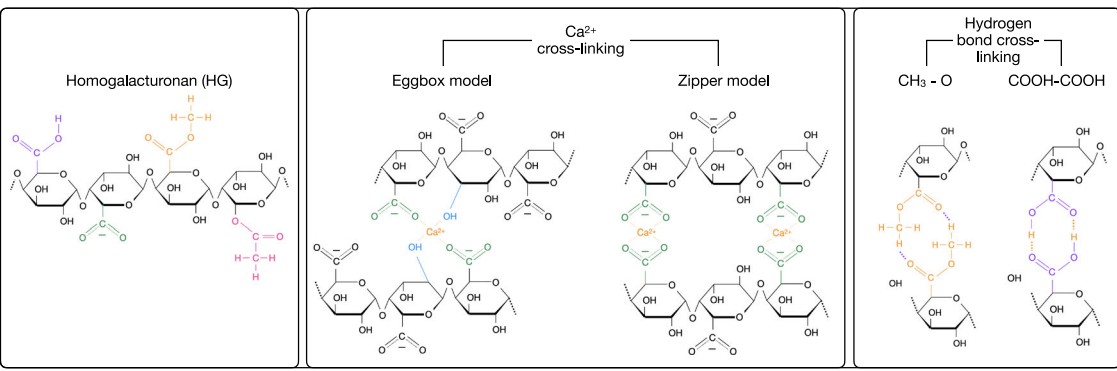

**Fig. 1 | Different cross-linking mechanisms for HG chains.** The different functional groups involved, namely protonated carboxyl groups (-COOH), de-protonated carboxyl groups (-COO⁻), methylated carboxyl groups (methyl-ester-ified GalA) and acetylated carboxyl groups, are highlighted in purple, green, orange and pink, respectively. Two distinct mechanisms have been proposed for the cross-linking of HG chains via Ca²⁺ bridges: the "egg-box" and the "zipper" model. The egg-box model is commonly found in the literature revolving around pectin cross-linking, and involves two hydroxyl (-OH, light blue) groups and two -COO⁻ groups. Here, we put forward the zipper model (which involves two -COO⁻ only) instead—in light of the computational results discussed in the "Results" section. HG cross-linking can also occur via hydrogen bonding, most prominently between methylated carboxyl groups - or even protonated carboxyl groups in very acidic conditions.

Specifically, we have investigated the molecular-level details of HG cross-linking via molecular simulations, quantifying the energetics of different cross-linking mechanisms. We have also systematically investigated the structure-function relationship between the functionalisation of HG chains and the kinetics of their aggregation. Our results suggests that the venerable "egg-box" model might not be adequate to describe the Ca²⁺-mediated cross-linking in pectin. To investigate the impact of pectin cross-linking on the porosity of the cell wall, we employed an extended version of the fluorescence microscopy-based technique pioneered by Liu et al.[21]. By visualizing changes in fluorescence from a fluorescently-labelled plasma membrane over time, we monitored the dynamics of a quenching molecule to penetrate the cell walls of wild type *Arabidopsis thaliana* plants and a mutant that is impaired in cell wall pectin cross-linking. In order to understand our experimental results, we used mathematical modelling based on fluid dynamics. Our results are consistent with the experimental data, provide a rationalisation of the time dependence of the fluorescence decay in terms of the morphology of the sample and lay the foundations for the future development of a predictive model - to quantify the porosity and potentially the tortuosity[22] of the cell wall.

## Results
### Molecular simulations
**The egg-box model of HG cross-linking is not supported.** As the most abundant pectic domain in plant cell walls is HG[23], we focused on understanding the interactions between HG pectin chains. Our aim was to elucidate how pectin cross-links at the molecular level—which in turn might help us to understand the different degrees of porosity within the cell wall. To do this, we investigated the energetics of each type of HG-HG interaction: the main options in this regard are summarised in Fig. 1. Firstly, we consider the carboxyl groups of GalA, which in most cases we would expect to find in their de-protonated form (-COO⁻), as their pKa is rather low (around 3.6). In this scenario, two -COO⁻ groups on two different chains can cross-link via a Ca²⁺ bridge.

Much of the existing literature suggests that these calcium bridges involve two hydroxyl (-OH) groups as well. This possibility, illustrated in Fig. 1, leads to the emergence of the so-called "egg-box" model for pectin cross-linking[14,24,25]. This model is widely accepted amongst plant scientists—however, it is difficult to resolve experimentally the exact position and—crucially—coordination of the Ca²⁺ ions within the cross-linking regions of HG chains[25].

Small-angle X-ray scattering (SAXS) has been previously employed to investigate the structural details of Ca²⁺ crosslinking in pectins and alginates[26,27]. However, it is important to note that SAXS cannot provide direct evidence for the egg-box model. Notably, even recent experimental work on arabidopsis[28] cites previous modelling efforts, specifically the work of Braccini et al.[29], as evidence for the egg-box model. Interestingly, Braccini et al.[29] do not advocate the original egg-box model but suggests a "shifted egg-box" configuration for galacturonate and guluronate chains.

In fact, recent work has called into question the validity of this model in favour of a "zipper" model[24], also illustrated in Fig. 1, which involves the two -COO⁻ groups only. This deceptively small difference between the egg-box and the zipper model does actually translate into a rather different configuration of the HG chain as a whole—as we will discuss in greater detail later.

In cases where the carboxyl group of GalA would be found in its protonated form, it is reasonable to expect some degree of hydrogen bonding between two -COOH groups belonging to different chains, without the need to involve a Ca²⁺ bridge[30]. In fact, whilst—in presence of a sufficient amount of calcium- the formation of -COO⁻ ‖ Ca²⁺ ‖ -COO⁻ bridges is the main mechanism of HG cross-linking, in highly methyl- and /or acetyl-esterified pectin the cross-linking takes places thanks to hydrogen bonding, as illustrated in Fig. 1.

Thus, the resulting porosity of the pectin network within the cell wall is determined by a competition of different HG cross-linking mechanisms. In order to assess the stability of these different cross-linking options for HG chains, we performed metadynamics simulations. The details are discussed at length in the "Methods" section: in short, metadynamics is an enhanced sampling computational technique that leverages molecular dynamics (MD) simulations to reconstruct the free energy profiles relative to one or more "collective variables"—in this case, specific degrees of freedom that describe the cross-linking process. To begin with, we have computed by means of unbiased simulations (i.e., without applying the metadynamics method so as not to tamper with the natural time evolution of the system) the running coordination number for Ca²⁺ in water (i.e. the average number of water molecules to be found within a given distance from the Ca²⁺ ion) so as to validate the reliability of our computational setup. The result, illustrated in Fig. 2a, indicates a rather extended first solvation shell (within 2.5 and 4 Å) that contains on average $6.5 \pm 0.5$ water molecules. This is consistent with previous results obtained via the CHARMM force field as well as first-principles MD simulations[31].

Via metadynamics simulations, we have investigated the free energy gain associated with the formation of a single -COO⁻ ‖ Ca²⁺ ‖ -COO⁻ bridge, which we report in Fig. 2b as a function of both the coordination number of Ca²⁺ with water and the coordination number of Ca²⁺ with any of the four oxygen atoms belonging to the two de-protonated carbonyl groups (see Fig. 2d). Our results indicate that Ca²⁺ is able to fully link with two -COO⁻ groups only via the "zipper" mechanism discussed above, without the need to involve -OH groups via the "egg-box" mechanism. The most stable link involves all the four oxygen atoms belonging to the two de-protonated

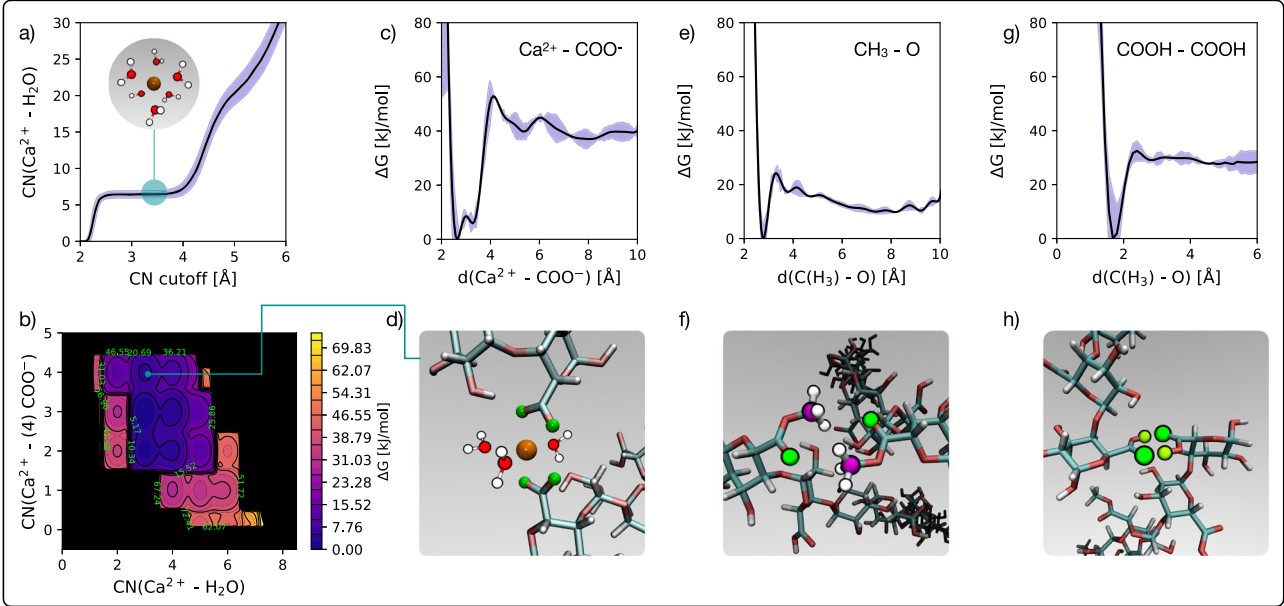

**Fig. 2 | Metadynamics simulations of HG cross-linking. a** Running average coordination number of Ca²⁺ with water molecules in an unbiased MD simulation. The shaded region highlights the uncertainty associated with our estimate (min–max error); the inset illustrated a six-coordinated Ca²⁺ ion. **b** Free energy surface relative to the formation of a single -COO⁻ ‖ Ca²⁺ ‖ -COO⁻ bridge, reported as a function of the coordination number of the Ca²⁺ ion with water molecules [CN(Ca²⁺ - H₂O)] as well as the coordination number of the Ca²⁺ ion with any of the four oxygen atoms belonging to the two -COO⁻ groups [CN(Ca²⁺ - (4)COO⁻)]. **c** Free energy profile relative to the interaction between a Ca²⁺ and a single COO⁻ group, reported as a function of the distance between the Ca²⁺ ion and any of the two oxygen atoms belonging to the COO⁻ group [d(Ca²⁺ - -COO⁻)]. **d** Representative configuration of a -COO⁻ ‖ Ca²⁺ ‖ -COO⁻ bridge. The Ca²⁺ ion and the four oxygen atoms belonging to the two -COO⁻ groups are coloured in orange and green, respectively. We also show the three water molecules the Ca²⁺ ion is coordinated to in that specific configuration. **e** Free energy profile relative to the interaction between two methylated carboxyl groups, reported as a function of the distance between one of the two carbon atoms within the -CH3 groups and one of the two oxygen atoms involved in the -C=O double bond; (**a**) typical configuration is depicted in (**f**), where the relevant carbon atoms are coloured in purple. **g** Free energy profile relative to the interaction between two carboxyl groups, reported as a function of the distance between one of the two de-protonated oxygen atoms and one of the two hydrogen atoms belonging to the carboxyl groups; a typical configuration is depicted in (**h**), where the relevant hydrogen atoms are coloured in lime.

carbonyl groups, whilst the coordination number of the Ca²⁺ ion with water becomes three (as opposed to the value of 6.5 observed in solution). In fact, in the unbiased simulations of HG cross-linking that we will discuss later, we have never encountered a situation resembling the "egg-box" mechanism. The interaction between -COO⁻ and Ca²⁺ alone is very strong, as illustrated by the free energy profile reported in Fig. 2c. Here, we look at the free energy gain in bringing a Ca²⁺ closer to the -COO⁻ group—in particular, the collective variable of choice is the distance between Ca²⁺ and either of the two oxygen atoms within the -COO⁻ group. The minimum of the free energy is characterised by ~50 kJ/mol, which is a value in excellent agreement with the previous estimate of the potential of mean force (a similar measure of energetics in this specific context) reported in the work of Assifaoui et al.[14].

In contrast, cross-linking of HG via hydrogen bonding seems to be leveraging much weaker interactions. This is quantified by the free energy profiles we have obtained relative to cross-linking via hydrogen bonding between either methyl-esterified GalA and -COOH groups, reported in Fig. 2e, g, respectively. The hydrogen bonding between -COOH groups is significantly stronger than that methylated carboxyl groups: ~30 kJ/mol versus ~20 kJ/mol. However, a single Ca²⁺ ‖ -COO⁻ interaction is much stronger than either of these hydrogen bonds. It is worth noticing that the -COOH ‖ -COOH can only happen in rather acidic conditions to begin with, and it is only reasonable to expect the interaction between two acetylated GalA to be even weaker than that between two methylated carboxyl groups.

Thus, in light of these results we argue that the main cross-linking mechanism for low methoxyl pectin in physiological conditions for plants must rely on -COO⁻ ‖ Ca²⁺ ‖ -COO⁻ bridges. In absence of sufficient Ca²⁺ and/or when dealing with high methoxyl pectin, hydrogen bonding interactions between methylated carboxyl groups might play a significant role,

but they are bound to result in less strongly linked pectin compared to the -COO⁻ ‖ Ca²⁺ ‖ -COO⁻ scenario. In turn, we argue that weaker HG-HG interactions are bound to result in more fluid pectin domains, thus ultimately leading to larger average pore size within the cell wall.

**Impact of HG functionalisation on its cross-linking potential.** To gain further insight into the kinetics of the HG cross-linking, we have also performed unbiased simulations of the aggregation of HG chains in water. In particular, we have considered HG chains containing either 8 or 40 GalA units, and we have systematically explored the impact of the functionalisation of the GalA units on the HG cross-linking. Our simulation boxes contained eight chains (either 8- or 40-unit long) featuring different combination of protonated -COOH groups (**P** or **p** for 8- and 40-unit long chains, respectively, in Table 1), de-protonated -COO⁻ groups (**D** or **d** in Table 1) and methylated carboxyl groups (**M** or **m** in Table 1). We summarise our results in Table 1, where we report the average number (computed over 40 ns-long MD simulations) of cross-links (via calcium bridges or via either -COOH ‖ -COOH or -CH3 ‖ -O hydrogen bonds) formed between the HG chains, the average lifetime of said cross-links and the average number of chains involved into the largest HG aggregate. We have verified the reproducibility of these results across 15, statistically independent MD simulations—as discussed in the Supplementary Information (SI; see Supplementary Table 1)

We find that D-only HG chains show the greatest extent of cross-linking, whilst P-only chains link much less effectively, and M-only chains even less so. This trend is not only evident in terms of the average number of cross-links, but the lifetime of the links is substantially longer for the -COO⁻ ‖ Ca²⁺ ‖ -COO⁻ bridges forming in D-only chains than for the hydrogen bonds formed in either P-only or M-only chains. These results are consistent with the energetics we have obtained by means of the

## Table 1 | HG cross-linking as a function of functionalisation

| System | $\langle Ca^{2+} \rangle$ | $\langle HB_{CH_3-O} \rangle$ | $\langle HB_{COOH} \rangle$ | $\langle \tau_{Ca^{2+}} \rangle$ | $\langle \tau_{HB_{CH_3}} \rangle$ | $\langle \tau_{HB_{COOH}} \rangle$ | $N_{agg}$ |
|---|---|---|---|---|---|---|---|
| PPPPPPPP | – | – | 9.91(32) | – | – | 1.43 | 6.03 |
| DDDDDDDD | 16.89 (32) | – | – | 8.02 | – | – | 8 |
| MMMMMMMM | – | 6.06 (32) | – | – | 1.65 | – | 3.94 |
| DDPDPPDP | 9.44 (16) | – | 4.23 (16) | 7.37 | – | 1.52 | 7.74 |
| DDMDMMDM | 8.66 (16) | 1.71 (16) | – | 7.88 | 1.53 | – | 7.35 |
| DPDPDPDP | 8.51 (16) | – | 6.59 (16) | 8.6 | – | 1.62 | 7.51 |
| DMDMDMDM | 9.09 (16) | 2.21 (16) | – | 5.53 | 1.97 | – | 6.98 |
| DDPPDDPP | 8.79 (16) | – | 4.72 (16) | 8.79 | – | 1.58 | 7.72 |
| DDMMDDMM | 8.46 (16) | 3.52 (16) | – | 8.43 | 1.57 | – | 7.54 |
| PPDDDDPP | 7.15 (16) | – | 3.56 (16) | 5.05 | – | 1.25 | 5.99 |
| MMDDDDMM | 8.16 (16) | 1.71 (16) | – | 7.11 | 1.33 | – | 4.81 |
| DDPPPPDD | 10.59 (16) | – | 3.88 (16) | 7.96 | – | 1.68 | 7.53 |
| DDMMMMDD | 9.49 (16) | 2.98 (16) | – | 6.35 | 2.09 | – | 7.92 |
| DPPDDPPD | 9.77 (16) | – | 4.09 (16) | 6.69 | – | 2.11 | 7.78 |
| DMMDDMMD | 9.77 (16) | 1.75 (16) | – | 8.62 | 1.37 | – | 7.38 |
| PDDPPDDP | 8.46 (16) | – | 4.25 (16) | 6.2 | – | 1.41 | 7.56 |
| MDDMMDDM | 7.95 (16) | 5.56 (16) | – | 8.96 | 1.89 | – | 7.63 |
| DDDDPPPP | 6.52 (16) | – | 3.41 (16) | 3.62 | – | 1.51 | 4.86 |
| DDDDMMMM | 7.80 (16) | 2.40 (16) | – | 5.41 | 1.66 | – | 3.85 |
| dddd | 58.05 (160) | – | – | 10.99 | – | – | 7.95 |
| mmmm | – | 1.81 (160) | – | – | 1.24 | – | 2.07 |
| dddp | 32.51 (120) | – | 2.79 (40) | 10.68 | – | 1.49 | 4.52 |
| dddm | 37.95 (120) | 0.17 (40) | – | 16.67 | 1.11 | – | 5.61 |
| ddpd | 36.12 (120) | – | 0.37 (40) | 12.41 | – | 1.04 | 6.07 |
| ddmd | 29.14 (120) | 0.98 (40) | – | 10.67 | 1.66 | – | 6.54 |
| ddpp | 32.07 (80) | – | 8.15 (80) | 11.93 | – | 1.94 | 5.14 |
| ddmm | 24.80 (80) | 0.50 (80) | – | 14.49 | 1.24 | – | 4.37 |
| dppd | 29.44 (80) | – | 6.27 (80) | 15.91 | – | 1.59 | 3.26 |
| dmmd | 28.85 (80) | 2.23 (80) | – | 11.00 | 1.90 | – | 3.44 |
| pddp | 21.24 (80) | – | 8.44 (80) | 11.06 | – | 1.69 | 3.17 |
| mddm | 16.83 (80) | 0.92 (80) | – | 9.80 | 1.63 | – | 2.63 |
| dpdp | 26.68 (80) | – | 10.37 (80) | 11.43 | – | 1.87 | 6.22 |
| dmdm | 31.44 (80) | 3.88 (80) | – | 14.63 | 1.61 | – | 5.39 |
| dppp | 15.29 (40) | – | 14.96 (120) | 10.28 | – | 1.82 | 3.45 |
| dmmm | 13.12 (40) | 1.73 (120) | – | 20.10 | 1.66 | – | 2.44 |
| pdpp | 12.24 (40) | – | 7.92 (120) | 14.53 | – | 1.45 | 1.93 |
| mdmm | 14.46 (40) | 2.04 (120) | – | 18.8 | 1.36 | – | 2.00 |

Our simulation boxes contained eight chains (either 8- or 40-unit long) featuring different combinations of protonated -COOH groups (**P** or **p** for 8- and 40-unit long chains, respectively), de-protonated -COO⁻ groups (**D** or **d** for 8- and 40-unit long chains, respectively) and methylated carboxyl groups (**M** or **m** for 8- and 40-unit long chains, respectively). The results have been computed by averaging over 40 ns-long MD simulations. We report: the average number of cross-links via calcium bridges, $\langle Ca^{2+} \rangle$; the average number of cross-links via CH₃-O hydrogen bonds, $\langle HB_{CH_3-O} \rangle$ (see Fig. 1); the average number of cross-links via -COOH hydrogen bonds, $\langle HB_{COOH} \rangle$ (see Fig. 1). We also report the average lifetime τ (in ns) of these cross-links as well as the average number of chains involved in the largest HG aggregate during the simulation, $N_{agg}$.

metadynamics simulations discussed above (and summarised in Fig. 2). In terms of the chains characterised by a mixture of either D and M or D and P units, it appears that 4-unit "blocks" of different units (e.g., DDDDMMMM or DDDDPPPP) cross-link to a much lesser extent if compared to chains with the same composition where however the units are alternating along the chains (e.g., DMDMDMDM or DPDPDPDP). This trend might indicate that alternating D units might favour hydrogen bonding between M or P units as well. However, it seems that chains terminated by D units (i.e., DDPPPPDD or DDMMMMDD) cross-link to an even greater extent than chains alternating D and either M or P units.

These results are consistent across 8-unit and 40-unit chains: interestingly, the lifetime of the -COO⁻ ‖ Ca²⁺ ‖ -COO⁻ bridges is on average almost two times longer than that we have computed for the 8-unit chains, whilst the lifetime relative to the hydrogen bonds remains largely unchanged. This is symptomatic of the collective effect realised by the formation of contiguous calcium bridges along the chains. In fact, our simulations allowed us to pinpoint quite clearly the occurrence of the "zipper" mechanism discussed above. In Fig. 3, we report a representative snapshot of the aggregation of eight D-only, 40-unit HG chains. Despite the relatively short duration of the simulation, we can clearly see the formation of calcium

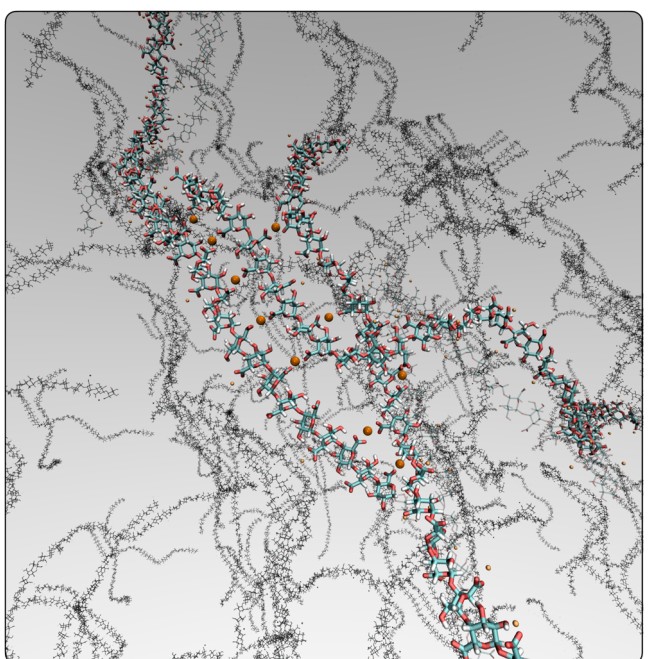

**Fig. 3 | Ca²⁺-mediated HG cross-linking.** A representative snapshot of an unbiased MD simulation of (8) HG chains, 40-unit each, taken after ~20 ns. Carbon, oxygen and hydrogen atoms within the three chains involved in the cross-linking are depicted in cyan, red and white, respectively. The Ca²⁺ ions involved in the cross-linking are depicted in orange. Note the emergence of the "zipper" mechanism (see text and Fig. 1).

bridges across multiple chains, without the need to involve any -OH group in the process (as advocated by the "egg-box" mechanism).

Overall, our simulations suggest that HG chains cross-link to each other according to this "zipper" mechanism that does not introduce significant torsion within the HG chain. In contrast, we note that for the -OH groups to be involved in the calcium bridges according to the "egg-box" model, the HG chains must distort to a significant extent. These considerations, whilst elucidating the molecular-level details of the cross-linking of HG chains, also suggest that the role of pH as well as the content of calcium within the cell wall might be key to determine the overall degree of cross-linking in pectin.

To better understand the influence of HG cross-linking on the porosity of the cell wall, we turn to the experimental measurements discussed in the next section. We remark at this stage that a large number of genes control pectin methyl esterification (and thus, have the potential to affect pectin cross-linking) and redundancy in their function means that genetic mutants are largely unaffected by loss of function of any single PME gene[16,17]. In contrast, mutants that are affected globally in HG pectin methyl esterification exhibit extreme structural defects, including reduced growth and loss of cell adhesion, due to their effects on the HG-rich middle lamella[25]. This makes it difficult to draw conclusions about cross-linking by comparing them with wild type plants.

To circumvent this issue whilst making appreciable alterations to pectin cross-linking, we have used a well-studied genetic mutant of *Arabidopsis thaliana* which exhibits qualitative differences in its RGII pectic domain but shows unaltered amounts of pectin and no defect in cell adhesion[32]. This mutation decreases the formation of boron bridges between pectin chains. RG-II is covalently linked to HG pectin[19] and in wild type plants, those boron bridges draw pectin chains closer together and facilitate further cross-linking via the mechanisms we have examined in our simulations[28,33]. As the formation of boron bridges and their knock-on effect on the creation of other linkages between pectin chains, (e.g., HG-Ca²⁺ bridges) are intertwined and cannot be disentangled experimentally, we focus on understanding their collective impact on the porosity of the cell wall.

## Experimental determination of the cell wall porosity in an arabidopsis cell-wall pectin cross-linking mutant

Recently, Liu et al.[21] presented a fluorescence microscopy-based technique that measures porosity of plant and fungal cells by assessing the ability of relatively large fluorescence quenching molecules to traverse the cell wall. Fluorescent labelling of the plasma membrane allows the degree of penetration of the quencher to be calculated on the basis of its ability to quench the fluorescence. This technique is an elegant refinement of previously used methods to assess effective pore size. To test the effect of altered pectin cross-linking on cell wall porosity, we applied this technique, with some modifications, to wild type arabidopsis Col0 plants and a well-studied mutant, *sensitive-to-freezing-8 (sfr8)*, which exhibits reduced pectin cross linking without exhibiting major developmental aberrations[34].

Central to our modification of the technique was the measurement of fluorescence over a series of time points, rather than through a single end-point measurement, to reveal the dynamics of the process. Specifically, we labelled the plasma membrane of epidermal leaf peels with the fluorescent dye FM4-64 and introduced a quencher molecule, trypan blue. We then monitored the decline of the observed florescence of the samples over a 15 min period. As can be seen from the images in Fig. 4a, fluorescence levels were similar between the two genotypes before quencher application but declined more rapidly in the *sfr8* mutant than in the wild type. Note that this difference was significant at the moment of application (time 0, $p$ value = 0.0141), and at time = 3 ($p$ value = 0.003) and 6 min ($p$ value = 0.0221) as well (see Fig. 4b). To investigate directly the pore size distributions in Col0 and *sfr8* mutant plants we prepared samples from Col0 and *sfr8* leaves for Scanning Electron Microscopy (SEM) measurements. Images (Fig. 4c) were analysed with FIJI to quantify both pore size (area), and the frequency of each pore size (Fig. 4d). Clear differences in pore size distribution between the two genotypes were observed. Specifically, the frequency distribution of the pore area reported in Fig. 4d indicates that *sfr8* plants have, on average, a larger proportion of larger pores (pore area $\gtrsim 70$ nm²) than the Col0 wild type and a smaller proportion of small pores (pore area $\lesssim 70$ nm²). These measurements allowed us to compute an estimate of the porosity $\phi$ of the samples as:

$$\phi = \frac{\langle A_p \rangle \cdot N_p}{A_s},\tag{1}$$

where $\langle A_p \rangle$ is the average pore size, $N_p$ is the number of pores and $A_s$ is the total area of the sample. Whilst the leaf to leaf variation in pore sizes varied noticeably (hence it was not appropriate to compare mean pore sizes directly across the two genotypes), we gauge $\phi$ to be of the order of $\approx 0.05$ for both samples, with $\phi_{\text{wild type}} < \phi_{\text{sfr8}}$. The outcomes of these experimental measurements were then used to construct a mathematical model (see next section) to rationalise the connection between the time-dependency of the observed fluorescence and the average pore size of the cell wall of different samples.

## Mathematical modelling of the porosity measurements

To gain further insight into the mechanistic aspects underpinning the experimental results reported in the previous section, we have constructed a minimalist mathematical model, based on fluid dynamics considerations. The variables and parameters used in this section are summarised in Table 2.

As illustrated in Fig. 5a, we consider a cell wall of area $A$ and thickness $h$. The cell wall sits in between the plasma membrane, with area $A$ (same as the area of the cell wall) and thickness $h_m$ (where $h_m \ll h$), and a bath of fluid containing the quencher molecule. The concentration of the quencher is defined as $q(z)$, where the $z$-coordinate runs perpendicular to the cell wall's surface, whilst the concentration of the quencher in the bath, $q_b$, is assumed to stay constant during the experiment. The concentration of the quencher in the membrane is assumed to be spatially homogeneous and as a function of time is defined as $q_m(t)$, with $q_m(0) = 0$ (i.e., the concentration of the quencher in the membrane at the start of the experiment is zero). The model assumes that the cell wall is structurally homogeneous (i.e., there is no dependence in $z$ on the properties of the wall), so that the concentration

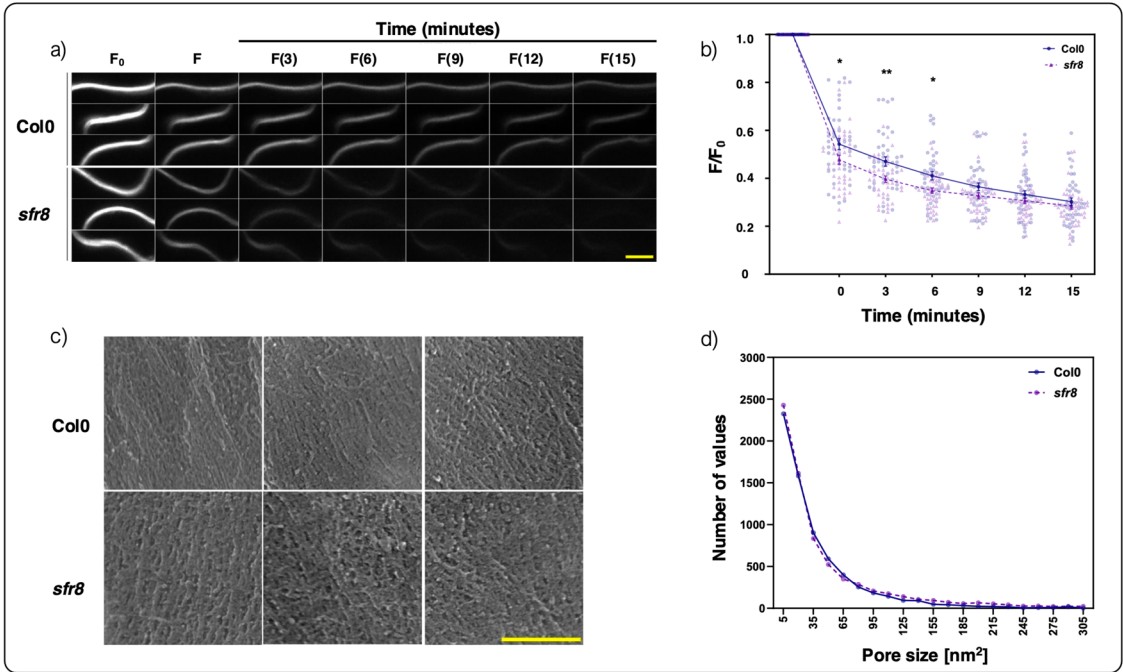

**Fig. 4 | Experimental measurements of the cell wall porosity in Col0 wild type and sfr8 *arabidopsis* leaves. a** Change in the fluorescence of the FM4-64-stained plasma membrane (see text) for Col0 and sfr8 cell wall samples. A time series of three representative fluorescence images reported for each point in time for each genotype are shown. The optical microscopy images were taken before quenching ($F_0$), immediately after addition of trypan blue (F) and at 3-min intervals thereafter. The scale bar (yellow) is 5 $\mu$m. **b** Change in the relative fluorescence ($F/F_0$) as a function of time. The mean values of $F/F_0$ are measured every 3 min within a 0–15 min time interval, for both Col0 and sfr8. The value at the Y-axis intersection represents the unquenched sample ($F/F_0$) and is equivalent to 1.0. Three separate Regions of Interest (ROI) were assessed per epidermal peel (39 and 42 ROI for Col0 and sfr8, respectively). The data were combined from three separate experiments. For statistical comparisons, a two-way ANOVA with Bonferroni's multiple comparisons test[82] was employed. The $p$ values at time points 0, 3, and 6 min are 0.0141, 0.003, and 0.0221, respectively. Other time points showed no significant difference. The "*" and "**" symbols indicate moderate ($p$ value $\ll 0.05$) and strong ($p$ value $\ll 0.01$) evidence, respectively, for a significant statistical difference between the two time series. **c** SEM images of Col0 and sfr8 cell walls. Three representative images, taken from 3 different leaves, are shown for both genotypes. The scale bar (yellow) is 0.25 $\mu$m. **d** Pore size (area, nm²) frequency distributions for Col0 and sfr8. The pore size distributions from the ROI (9 and 11 ROI for Col0 and sfr8, respectively) of three individual plants per genotype was assessed. A frequency distribution with a bin width and range of 15 and 5–305 nm², respectively, was employed.

gradient across it is:

$$\frac{dq}{dz} = (q_b - q_m(t))/h \, . \tag{2}$$

The cell wall is assumed to be a porous medium, characterised by porosity $\phi$ (i.e., the ratio between the volume of empty space within the cell wall and the total volume of the cell wall) and tortuosity $\eta(\phi)$. The latter is a measure of the complexity of the pathway for the fluid (and thus, in our case, the quencher) to diffuse through the porous network of the cell wall. The quencher molecules are characterised by a diffusion coefficient $D$ in the bath. The values of $D$ reported in Table 2 have been estimated by means of MD simulations of a single quencher molecule in water at room temperature and ambient pressure (see "Methods" section). The diffusion coefficient of the quencher through the pores of the cell wall, however, is bound to be (much) lower than the diffusion coefficient in solution. As such, we assume that within a pore the quencher has diffusion coefficient $CD$, where $C \ll 1$ (ref. 35). Here, we write down the "effective" diffusion coefficient $D_e$ of the quencher in the cell wall as[36]:

$$D_e = \frac{C\phi D}{\eta(\phi)} \, . \tag{3}$$

We also assume that the quencher follows Fickian diffusion:

$$J_D = -D_e \frac{\partial q}{\partial z}, \tag{4}$$

where $J_D$ is the diffusional flux (the amount of quencher diffusing in the $z$ direction per unit area per unit time). At time $t$, the amount of quencher diffusing per unit time *into* the cell wall is $-AJ_D(t)$. Thus, the total amount diffused by time $t$ is $Q(t) = \int_0^t -AJ_d(s)ds$. This is diffusing into the membrane, and relating this to the concentration gives $Q(t) = Ah_m q_m(t)$, and also $\frac{dQ}{dt} = -AJ_d(t)$.

Combining these equations, we get the ordinary differential equation:

$$\frac{dq_m}{dt} = -\frac{1}{h_m} J_d(t) = \frac{D_e}{h_m h}(q_b - q_m(t)) \, , \tag{5}$$

with solution

$$q_m = q_b \left( 1 - \exp\left(\frac{-D_e}{h_m h}t\right) \right). \tag{6}$$

This has characteristic timescale

$$T = \frac{h_m h}{D_e} = \frac{h_m h \eta(\phi)}{\phi CD} \, . \tag{7}$$

Having found the concentration of the quencher in the membrane, we must now relate this to the fluorescence as measured in the experiments. The fluorescence $F$ at $t = 0$ is denoted by $F(0) = F_0$, and the relative intensity defined as $I = F/F_0$. This is related to the quencher concentration by the

**Table 2 | Variables and parameters utilised in the mathematical modelling of the porosity measurements**

| Name | Symbol | Units | Reference Value |
|---|---|---|---|
| Time | $t$ | s | |
| Concentration of quencher in membrane | $q_m(t)$ | mol m$^{-3}$ | |
| Diffusional flux | $J_D(t)$ | mol m$^{-2}$ s$^{-1}$ | |
| Fluorescence intensity | $F(t)$ | – | |
| Porosity | $\phi$ | – | |
| Tortuosity | $\eta(\phi)$ | – | $1/\sqrt{\phi}$ |
| Quencher rate coefficient in membrane | $k_q$ | m$^3$ mol$^{-1}$ s$^{-1}$ | ? |
| Decay timescale | $\tau_0$ | s | ? |
| Quencher constant | $K = k_q \tau_0$ | m$^3$ mol$^{-1}$ | ~10$^4$ |
| Diffusion coefficient of quencher in bath | $D$ | m$^2$ s$^{-1}$ | $0.56 \times 10^{-4}$ |
| Diffusion coefficient of quencher in pore | $D_p = CD$ | m$^2$ s$^{-1}$ | $C \ll 1$ |
| Wall thickness | $h$ | m | ~$280 \times 10^{-9}$ |
| Membrane thickness | $h_m$ | m | ~$10 \times 10^{-9}$ |
| Concentration of quencher in bath | $q_b$ | mol m$^{-3}$ | $0.5 \times 10^{-3}$ |
| Sample area | $A$ | m$^2$ | known |

Stern–Volmer equation[37]:

$$I = \frac{1}{(1 + Kq_m)}, \tag{8}$$

which with (6) rearranges to:

$$K\left(1 - \exp(-t/T)\right) = (1/I - 1)/q_b \tag{9}$$

This is the function used for the fitting of the experimental data reported in Fig. 5. We fit $I$ as a function of $t$, which gives us values for $K$ and $T$ for each plant type.

In our case, from this fit we have obtained values for $T_{\text{Col0}}$ and $T_{\text{sfr8}}$. If we assume that the porosity (and possibly the tortuosity) are the only differences between the two samples, then:

$$\frac{T_{\text{Col0}}}{T_{\text{sfr8}}} = \frac{\eta_{\text{Col0}} \phi_{\text{sfr8}}}{\eta_{\text{sfr8}} \phi_{\text{Col0}}}. \tag{10}$$

To relate the tortuosity to the porosity, we use the simple Bruggeman expression[38], $\eta = \phi^{-1/2}$. This assumes that the porous structure is homogeneous and isotropic, and formed of particles much smaller than the size of the sample. Using this, we obtain:

$$\frac{T_{\text{Col0}}}{T_{\text{sfr8}}} = \frac{\phi_{\text{Col0}}^{-1/2} \phi_{\text{sfr8}}}{\phi_{\text{sfr8}}^{-1/2} \phi_{\text{Col0}}} = \frac{\phi_{\text{sfr8}}^{3/2}}{\phi_{\text{Col0}}^{3/2}} = \left(\frac{\phi_{\text{sfr8}}}{\phi_{\text{Col0}}}\right)^{3/2}, \tag{11}$$

so that:

$$\frac{\phi_{\text{sfr8}}}{\phi_{\text{Col0}}} = \left(\frac{T_{\text{Col0}}}{T_{\text{sfr8}}}\right)^{2/3}. \tag{12}$$

From the fit reported in Fig. 5 we get:

$$\frac{\phi_{\text{sfr8}}}{\phi_{\text{Col0}}} = 1.4717. \tag{13}$$

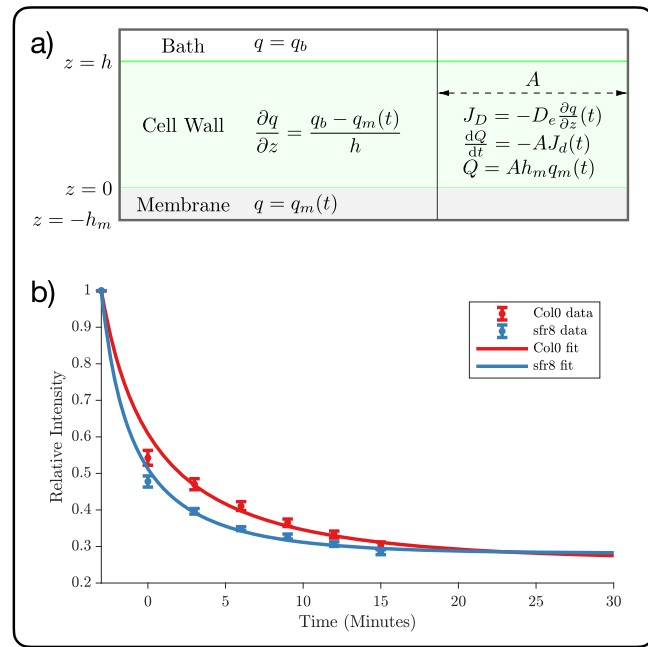

**Fig. 5 | Mathematical modelling of the porosity measurements. a** Schematics of the model. The relevant variables and parameters are reported in Table 2. **b** Fitting of the experimental data (see Fig. 4).

This result indicates that the cell wall of the *sfr8* mutant is ≈1.5 more porous than the cell wall of the wild type. However, we stress that in order for this model to be able to provide quantitative prediction of the porosity and—crucially—the tortuosity of the cell wall of each sample, further experimental efforts are needed, aimed at, e.g., "calibrating" the model via measuring the time dependence of the fluorescence for samples of known porosity and/or tortuosity.

Nevertheless, despite the simplicity of our model, our results indicate that: (i.) the experimental results can be explained without the need of introducing any "blocking" mechanism for the quencher to get stuck in and/or occlude the pores of the cell wall[39]; (ii.) measuring the decay in the fluorescence of the sample as early as possible during the experiments is justified by the fact that by doing so one maximises the impact of the different porosity of the samples; (iii.) the different porosity and/or tortuosity of the samples alone is sufficient to explain the difference in relative intensity observed experimentally. Future work from the modelling standpoint might explore more sophisticated models for the tortuosity, as this parameter provides information about the morphology of the pore network.

## Discussion

The plant cell wall is a highly complex structure governed by numerous interactions between carbohydrate polymers and structural proteins including extensins[40]. These interactions, particularly the formation of cross-links between pectin chains, determine not only the physical strength of the structure but its porosity, which impacts upon the ease of entry of a range of small, biologically important macromolecules, including enzymes, cell wall structural proteins, intercellular signalling molecules and carbohydrates[41].

To gain a better understanding of what shapes porosity in plant cell walls, we began with the pectin building blocks and examined the possible molecular interactions between pectin chains that are capable of cross-linking pectin and making it less porous. Our MD simulations added evidence to an emerging case to be made for a zipper model rather than the popular egg box model favoured by many plant biologists. Interestingly, our MD simulations also highlighted the importance of pH on the wall and

cross-linking. These data are consistent with experimental observations in vitro that pectin gelling and firmness is dependent upon both $Ca^{2+}$ and pH[42,43]. In fact, the acidity of the wall has long been appreciated as determining growth and extensibility of the wall for a variety of reasons, which include the pH dependence of $Ca^{2+}$ cross-linking of HG[44].

Having investigated the molecular basis for the different ways pectin chains can aggregate and interact with one another, we gathered experimental data from a biological system in which pectin cross-linking is altered. We used time-dependent fluorescence quenching measurements to assess how altered cross-linking affected the porosity of cell wall. We confirmed the validity of these measurements with direct observations via SEM and we modelled the fluorescence quenching data as well. Whilst arabidopsis mutants in HG pectin methyltransferases are available and would be predicted to exhibit altered crosslinking, these suffer from numerous defects including stunted growth and a loss of cell adhesion[45–47]. Thus, to obtain the experimental data for our porosity modelling, we used a more amenable arabidopsis mutant, which shows no cell adhesion defects and minimal effects on growth[32]. *Sensitive-to-freezing-8* (*sfr8*)[34] harbours a mutation in the well-studied *mur1* gene, known to influence RGII pectin cross-linking[48]. Due to a modification of the sugar side chain A, RGII pectin cross-linking is reduced in *sfr8/mur1* mutants. In wild type arabidopsis cell walls, RGII pectic domains can cross-link via borate diester linkages. RGII is covalently linked to HG pectin[33] so in turn, these linkages bring the HG chains closer to each other, thus resulting in an enhanced probability for HG chains to further cross-link to each other. This is borne out of experimental observations that demonstrate the dependency of HG-$Ca^{2+}$ cross-linking on RGII boron bridges. Pectin was more vulnerable to EDTA extraction in *mur1* mutants, which exhibit relocation of non-methylesterified HG[28]. Furthermore, boron-bridged RG-II and calcium are both required to maintain the pectin network of the arabidopsis seed mucilage ultrastructure[28] and affect pectin properties including viscosity. The marked differences in porosity observed in cell walls with and without boron cross-linking of this relatively minor pectic domain[5] would support the idea that reducing RGII boron bridges has a downstream and greater widespread effect on HG cross-links also.

To investigate this hypothesis, we used a modified version of the fluorescence quenching method developed by Liu et al.[21], to probe the porosity of wild type arabidopsis and *sfr8* mutant cell walls. As expected, our results showed that the quenching molecule traversed the cell wall to quench the fluorescence in the membrane more quickly in *sfr8* than in Col0 wild type, consistent with reduced cross-linking. To validate our results, we used SEM to image the cell wall matrices of leave and assess the amount of visible pore space are in each. Consistent with our fluorescence results, this confirmed that in the mutant, the distribution of size areas is altered: *sfr8* exhibits fewer smaller pore areas and a greater number of larger pore areas.

In order to rationalise the experimental fluorescence measurements, and particularly to elucidate the physical processes underpinning the results, we have built a straightforward mathematical model leveraging fluid dynamics concepts. Our results suggests that the flux of the quencher molecules through the cell wall does not occlude the pores within the pectin network; although in future, modifications to this model which include a "blocking mechanism"[39] could be considered when dealing with larger quencher molecules.

We also provide a justification for the up-to-now empirical choice of measuring the fluorescence decay within the sample as soon as possible after having introduced the quencher. This is because the early time regimes maximise the difference in terms of quenching efficiency. Finally, we demonstrate that the different responses in terms of fluorescence elicited by different samples can be explained in terms of the porosity of the samples alone.

The ability to predict the interactions between pectin chains in different contexts and forecast changes in porosity has many applications in agri-technology. Pectin structure determines the level of success attacking fungal and bacterial pathogens may have[8,49,50] and mechanisms have even been identified whereby the invading fungus targets Ca-pectate domains to

render the wall more porous and facilitate access to the cell's nutrients[51]. Growing evidence suggests that remodelling of cell wall and particularly pectin structure is involved in the defence against abiotic stresses[52–54] and there are reports of wall porosity affecting susceptibility to ion toxicity and dehydration[50,55,56]. Altering the porosity of cell wall has also been shown to increase the efficiency of $CO_2$ uptake, increasing photosynthesis as a result[57]. In addition, differences in cell wall porosity can be exploited in optimization of nanoparticle uptake[58], wood quality and strength[59] and saccharification of plant and algal material to produce biofuels[10,60,61]. There are clearly opportunities for crop protection here.

In summary, this study shed light on the intricate dynamics of pectin cross-linking and its consequential effects on plant cell wall porosity, offering significant insights across multiple disciplines. By elucidating the molecular interactions within pectin chains and their impact on cell wall structure, we have advanced our understanding of plant biology, particularly in relation to plant growth, disease resistance, and stress responses. Our findings highlight the potential for manipulating pectin structure to enhance crop resilience against pathogens and abiotic stress or improve saccharification potential, pointing to novel strategies for crop protection and improvement. This cross-disciplinary research not only deepens our fundamental understanding of plant cell wall chemistry but also opens new avenues for agricultural innovation and sustainability.

## Methods
### Growth of plants
Arabidopsis plants were grown under 12 h light: 12h dark at 20 °C ± 1 °C, 100–150 $\mu$mol/s/m$^2$ in a walk-in growth room as described previously[34]. *sensitive-to-freezing-8* (*sfr8*)[34] and Columbia-0 (Col0) wild type plants were compared. Plants were grown to the rosette leaf stage (approximately 5 weeks post germination). Leaves were removed for porosity measurement or analysis by scanning electron microscopy.

### Porosity measurements
**Fluorescence measurements.** The porosity of the cell wall in leaf epidermal peels was assessed through a fluorescence quenching assay, following the method originally described by Liu et al.[21]. We used the detailed protocol outlined for maize leaves in the work of Liu et al.[62] with the following modifications. Abaxial epidermal peels were obtained from 5-week-old plants using the Perforated-tape Epidermal Detachment (PED) technique described by Ibata et al.[63]. The peels were incubated for 5 mins in a 20 $\mu$M FM4-64 (Thermo Fisher Scientific) solution. Following incubation, the epidermal peel was rinsed with water and then mounted on a microscope glass slide. Before mounting, Scotch Tape was carefully applied along the long edges of the glass slide, creating a chamber in the centre where the sample was placed and covered with a cover slip. An initial image ($F_0$) was captured prior to the addition of 0.5 $\mu$M trypan blue (Sigma-Aldrich) to quench the fluorescence of the sample. Subsequently, a series of images ($F$) were acquired at 3-min intervals over a span of 15 mins. Imaging was carried out using a Zeiss 800 laser scanning confocal microscope equipped with a 1.4 NA 63x oil immersion objective, using a 488 nm excitation laser and fluorescence detection at 546–618 nm. The ZEN Blue imaging software provided by the microscope manufacturers (Zeiss) was utilized for image acquisition, while analysis was done using ImageJ software[64]. For image analysis, the mean intensity of a defined region of interest (ROI), specifically a 15 $\mu$m x 5.6 $\mu$m rectangle, was quantified. Three separate ROIs were assessed for each sample. Intensity values were exported, and the relative porosity calculated as $F/F_0$.

**Electron microscopy.** Chemicals were obtained from Sigma, Poole, UK or Fisher Scientific (unless otherwise stated). A single leaf from a 5-week old arabidopsis plant was placed in PEM buffer[65] (25 mM PIPES [1,4-piperazine-diethanesulfonic acid], 0.5 mM $MgSO_4$, 2.5 mM EGTA, pH = 7.2) at room temperature and cut to a width of 1–2 mm in the buffer. The tissue sample

was transferred to PEM buffer with 4% formaldehyde (TAAB, Aldermaston, UK) and placed on ice for 15–30 min. Samples were then transferred to Karnovsky's fix (3% glutaraldehyde (TAAB), 2% formaldehyde in PEM buffer) and incubated overnight at 4 °C. The following day, samples were washed 3–5 times for 5–10 min in PEM buffer, before incubating in 0.5% sodium hypochlorite in PEM buffer for 20 mins at room temperature. After five 10-min rinses in PEM buffer, samples were transferred to 2% osmium tetroxide (Agar Scientific Ltd., Stansted, UK) in PEM buffer, for 60–120 min at 4 °C. Samples were dehydrated using an ethanol series with 5–10 mins at 4 °C in each of the following: 30%, 50%, 70%, 80%, 90%, 95% ethanol. Dehydration was completed using 3 successive washes in 100% ethanol for 10 min each at room temperature. After critical point drying (CPD) in a Leica Critical Point Dryer EM CPD030, the sample was attached to a piece of carbon double-sided adhesive tape (aluminium base; NISSHIN-EM CO., LTD) attached to a 5-mm × 5-mm silicon mount (Agar Scientific). The sample was then surrounded by silver paint (Micro to Nano, Haarlem, Netherlands) to improve the conductivity between the sample and the silicon wafer and sputter coated with approximately 2 nm of platinum at a 45 degree angle, using a 328UHR coating unit (Cressington Scientific instruments, Watford, UK). Pore size was determined from leaves obtained from three individual plants for Col0 and four individual plants for *sfr8*, utilizing the methodology outlined by Hojat et al.[66]. In brief, SEM images were processed using ImageJ software[64], where a region of 750 nm × 500 nm was selected from each image for analysis. Initially, thresholding was applied to segment the images into pore and non-pore regions, followed by the application of morphological filters, erode and dilate. The size and quantity of pores were extracted from each image and converted to nanometres. Pore areas falling within the range of 5–300 nm$^2$ were utilized to generate a frequency distribution graph, while those outside this range were excluded from the analysis.

## Molecular dynamics simulations

Molecular dynamics simulations were performed in GROMACS 5.1.3[67], using the all-atom CHARMM36 forcefield[68] together with either the TIP4P water model[69] or the TIP4P/Ice water model[70]. Initial simulations were carried out using TIP4P/Ice, as this model has been shown to be particularly effective for simulations of biomolecules in supercooled water[71–73]. Due to the dynamics of these systems being extremely slow in TIP4P/Ice, and therefore difficult to explore on the simulation timescale, we have chosen to switch to the TIP4P water model, which at room temperature is characterised by a much faster mobility. This choice allowed us to investigate the aggregation of the HG chains without the need to resort to enhanced sampling simulations - too computationally expensive to deal with the systematic investigation we have conducted here. Additional details with respect to this computational strategy can be found in the SI, Supplementary Fig. S1.

Simulations have been run on a range of systems with different HG chain compositions (see Table 1). Each system consisted of eight identical chains with eight residues each, which were either protonated (P), deprotonated (D), or methylesterified (M). Deprotonated residues carry a negative ($1^-$) charge, so an appropriate number of calcium ($2^+$) ions were added to systems containing chains with deprotonated residues to balance the overall charge. The protocol for these simulations was as follows. The HG chains were placed randomly into a cubic simulation box of initial side 10 nm. The system was then solvated in water, and if necessary, some water molecules were replaced with calcium ions to balance the charge. An energy minimisation was then carried out using a steepest descent algorithm[74], followed by a 60 ns run, with a timestep of 2 fs, at room temperature and ambient pressure in the NPT ensemble. The Bussi-Donadio-Parrinello thermostat[75] and Berendsen barostat[76] were used, with coupling constants of 0.5 and 4 ps, respectively. Periodic boundary conditions were applied in three dimensions. Additional computational details with respect to these MD simulations can be found in the SI, Supplementary Table 2.

To investigate in detail the energetics of the different interactions illustrated in Fig. 1, we have resorted to well-tempered metadynamics

simulations[77,78] using PLUMED 2.4.2[79]. Well-tempered metadynamics is an enhanced sampling technique that allows the free energy surface of system to be explored relative to a chosen set of degrees of freedom, or "collective variables" (CVs). These simulations have been used to measure the relative strengths of the two HG linking methods, namely calcium bridges and hydrogen bonds. The simulations contained two identical HG chains of eight residues, with seven protonated residues and either one deprotonated residue or one methylesterified residue. For the calcium bridge, the distance between the calcium ion and the deprotonated site was chosen as the CV to bias: the width, height and deposition stride of the Gaussian potentials were set (after extensive testing and validation) to $\sigma = 0.04$ nm, $W = 1.4$ kJ/mol, and 500 steps, respectively. The bias factor was set to $\gamma = 60$. For the hydrogen bond, the distance between the methyl carbon on one residue and the carbonyl oxygen on the other residue was chosen as the CV to bias: the width, height and deposition stride of the Gaussian potentials were set (after extensive testing and validation) to $\sigma = 0.05$ nm, $W = 1.2$ kJ/mol, and 500 steps, respectively. The bias factor was set to $\gamma = 60$. Additional computational details with respect to these metadynamics simulations can be found in the SI, Supplementary Table 3.

The deposition rate of the bias was set to 500 MD steps (equivalent to 1 ps) as the structural relaxation time of TIP4P water at room temperature and pressure is of the order of $10^{-1}$ ps[80]. The width of the Gaussian potentials was initially set to $\sigma = 0.01$ nm. This value turned out to be too small, in that the resulting free energy surfaces struggled to achieve convergence due to the too-fine resolution. Thus, we have increased $\sigma$ to 0.04 and 0.05 nm for investigating the calcium bridge and the hydrogen bond interactions, respectively. Values of $\sigma > 0.08$ nm resulted in free energy surfaces lacking sufficient detail/resolution. $W$ was initially set to half the value of the thermal energy ($k_B T$) at room temperature, i.e., 1.24 kJ/mol. This is a common rule of thumb in terms of striking a compromise between sampling efficiency and accuracy. We have explored values of $W$ between 1.00 and 2.00 kJ/mol, in conjunction with bias factors (which value also influences said efficiency/accuracy ratio) ranging from 50 to 100. Any combination of the values of $W$ and $\gamma$ within the above-mentioned ranges led us to obtain free energy surfaces within the uncertainty computed via the reweighing technique of Tiwary and Parrinello[81] (and reported in Fig. 2 as the shaded blue regions) within an acceptable simulation time (500–700 ns).

Whilst the concept of time in metadynamics simulations does not have a direct physical meaning (as the time evolution of the system is heavily influenced by the artificial bias introduced for the purposes of exploring the free energy surface of interest), such long simulation times were necessary to ensure the proper convergence of the resulting free energy surface. To this end, we have: (i.) monitored the changes of the free energy surface as a function of the simulation time; (ii.) explored the impact of varying the height and width of the bias potential; (iii.) utilised the reweighing technique of Tiwary and Parrinello[81] to provide a quantitative estimate of the uncertainty associated with the free energy surface.

**Statistics and Reproducibility.** We have included a discussion of the reproducibility of our results with respect to every aspect of our methodology (see the Methods section). Perhaps the most delicate aspect of our work concerns the reproducibility of our results in terms of the molecular dynamics simulations of HG aggregates (see Table 1). To this end, we have included direct evidence of the robustness of these results in the Supplementary Information. With respect to the fluorescence measurements reported in Fig. 4b, three separate regions of interest (ROI) were assessed per epidermal peel (for a total of 39 and 42 ROI for Col0 and *sfr8*, respectively). The data were combined from three separate experiments. For statistical comparisons, a two-way ANOVA with Bonferroni's multiple comparisons test[82] was employed. Significance was set at $\alpha = 0.05$. In terms of the SEM measurements, the pore size distributions from the ROI (9 and 11 ROI for Col0 and *sfr8*, respectively) of three individual plants per genotype was assessed. A frequency distribution with a bin width and range of 15 and 5–305 nm$^2$, respectively, was employed. Analyses were performed using GraphPad Prism version 10.1.0 for Windows. In terms of the results of the metadynamics

simulations reported in Fig. 2, we have utilized the reweighing technique of Tiwary and Parrinello[81] to assess the uncertainty associated with the resulting free energy profiles.

## Reporting summary

Further information on research design is available in the Nature Portfolio Reporting Summary linked to this article.

## Data availability

All of the data associated with this study are available via a publicly accessible GitHub repository (PEC_LINK, https://github.com/gcsosso/PEC_LINK.git). We have also linked a release of said repository to a Zenodo repository (also publicly accessible), for which we have obtained a permanent DOI[83]. Any remaining information can be obtained from the corresponding author upon reasonable request.

## Code availability

The details of the codes utilized in this work can be found in the Methods section. No in-house or proprietary code has been utilized. With respect to the computational aspects of the work, we remark that both the GROMACS and the PLUMED packages are open source and publicly available.

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

## Acknowledgements
This work has been supported by the Biotechnology and Biological Sciences Research Council (BBSRC) grant "Understanding ice formation in plants: finding new routes to freezing tolerance (PlantIce)" (BB/V015559/1). N.R. was funded by a BBSRC Doctoral Training Partnership studentship (BB/ M011186/1). P.L.J. is funded via the EPSRC CDT in Modelling of Heterogeneous Systems (HetSys, grant number EP/S022848/1). J.E.S acknowledges support from the EPSRC under grants EP/W031426/1, EP/S029966/1 and EP/P031684/1. I.J.P., F.B and G.C.S. gratefully acknowledge the usage of the ARCHER2 UK National Supercomputing Service (https://www.archer2.ac.uk), which we have accessed via the HecBioSim consortium (project e676), funded by the EPSRC (Grant No. EP/X035603/1). I.J.P. and G.C.S. also gratefully acknowledge the use of SULIS, which was funded by the EPSRC (EP/T022108/1), via the HPC Midlands+ Consortium. I.P., F.B. and G.C.S. would also like to acknowledge the high-performance computing facilities provided by the Scientific Computing Research Technology Platform (SCRTP) at the University of Warwick. H.K, T.J.H. and I.O. would like to acknowledge Andrew Iskauskas for crucial/helpful discussions on statistical analysis of SEM measurements.

## Author contributions
G.C.S., H.K, T.J.H., I.O. and I.J.P. conceived the research. I.J.P., F.B. and G.C.S. performed and analysed the molecular dynamics and metadynamics simulations. P.L.J., G.C.S. and J.E.S. devised applied and analysed the fluid dynamics-based mathematical model. I.O., T.J.H., H.K. and M.W.G. devised the experimental studies. I.O. and N.R. extended the methodology and performed the fluorescence-based porosity assays. C.K.I. developed the SEM methodology and performed, together with I.O., the EM measurements. I.O., I.J.P., P.L.J., F.B., T.J.H., J.E.S., H.K. and G.C.S. contributed to writing the manuscript.

## Competing interests
The authors declare no competing interests.
