## [Transparent Peer Review file · Communications Biology]

Understanding Pectin Cross-linking in Plant Cell Walls

Corresponding Author: Professor Gabriele Sosso

Version 0:

Reviewer comments:

Reviewer #1

(Remarks to the Author)

Obomighie et al, performed an interesting piece of work aimed enhancing our understanding of the mechanisms of pectin cross-linking and its impact on cell wall porosity using a combination of combining molecular dynamic simulations, mathematical modelling as well as experimental investigations. They provide an excellent background into the problem and chosen very advanced and sound techniques to address the problem ultimately provide strong evidence favouring a zipper model for pectin cross-linking. My only concern relates to the experimental biology side of the work where most of my expertise lies

1. With regards to the aim of understanding the influence of HG cross-linking on porosity of the cell wall (page 9) and also given the novelty of the observations and claims herein, it may help to actually run simulations or perform experiments in Arabidopsis plants with altered levels of pectin methyl esterification (similar to the plants used in the study by Weraduwage et al., 2016 (more details below)). The authors have indicated that redundancy of methyl esterification genes could be a limiting factor but I am sure it will be helpful to at least see what happens even with small changes in the methyl esterification status of the wall.

a. Details of the reference: Sarathi M Weraduwage 1 , Sang-Jin Kim 1 , Luciana Renna 1 , Fransisca C Anozie 2 , Thomas D Sharkey 2 , Federica Brandizzi 2 Plant Physiol. 2016 Jun;171(2):833-48.doi: 10.1104/pp.16.00173. Epub 2016 Apr 4. Pectin Methyl esterification Impacts the Relationship between Photosynthesis and Plant Growth. PMID: 27208234 PMCID: PMC4902601 DOI: 10.1104/pp.16.00173

2. Perhaps not a very big deal, but could the authors explain why the quenching level for the lower sfr8 mutant was similar to Col(0), ie will this more frequently if they tested more samples and if so what could possibly be the reason

3. For future reviews it may also be helpful for authors to number text lines to help with referencing

Reviewer #2

(Remarks to the Author)

This study utilized a multidisciplinary approach including in silico approaches and experimental testing to provide novel insights into the mechanism of cross-linking of pectin and revealed a strong link between HG cross-linking and porosity of the cell wall. The work is intriguing, especially in the context of our new understanding of the plant primary cell wall model, and the supporting evidence are illustrative and engaging. Overall, the research quality is good, there is sufficient data presented in the results and discussed afterwards. However, additional editing is needed to enhance quality. The following are the three most important improvements that the author needs to make.

1.The Arabidopsis cell wall mutant sensitive-to-freezing-8 (sfr8) in which RGII pectin cross-linking is reduced was selected for experimental validation. The work indicated that disrupted RGII boron bridges affected HG-Ca²⁺ crosslinking, which further influenced pectin properties including cell wall porosity. HG is the most abundant pectin subtype in plant cell wall, while RGII only makes up a small percentage of pectin. I was wondering why not use cell wall mutants where HG cross-linking is directly disrupted for experimental validation? Can this point be explained in greater detail?

2.In the section of "Conclusions", the authors mentioned that predicting the interactions between pectin chains in different contexts and forecasting changes in porosity has many applications in agritechology, followed by giving several examples from previous studies. However, these references cited by the authors seem too general. I suggest cite several more updated references which can specifically reveal a strong relationship between cell wall porosity and stress tolerance etc.

3.The authors made a conclusion that this study challenge the prevailing egg-box model, favoring a zipper model for pectin cross-linking instead. The conclusion was made only based on molecular simulations. I was wondering if any experimental data can be provided to support this conclusion.

Reviewer #3

(Remarks to the Author)

The manuscript by Obomighie et al. used a multidisciplinary approach spacing from MD simulations to experimental assays to clarify the cross-linking of Pectin and the relationship between its structure and the porosity of the cell wall. This paper is well written and the conclusions are supported by their analyses, thus representing a step forward in understanding the molecular basis of such a mechanism. I can recommend the publication of the manuscript after the following points will be addressed:

- 1- Can the authors provide a brief description in the SI of the initial simulations they run using the TIP4P/Ice model and a quantitative explanation of why they switched to the TIP4P one?
- 2- The authors should provide more details about the metadynamics simulations: i) what's the length of such simulations? ii) they should provide a better explanation of why they finally used those specific values of width, height and deposition stride of the Gaussian potentials; iii) are they sure that the convergence was reached?

Version 1:

Reviewer comments:

Reviewer #1

(Remarks to the Author)

Dear Editor,

Thus far, the authors have addressed all the points I raised. This includes both experimental and general questions.

Thank you

Reviewer #3

(Remarks to the Author)

The authors addressed all my comments. Thus, I can recommend its publication.

Please note that all the changes discussed in this rebuttal are **highlighted in red** within the marked-up version of the revised manuscript. Please also note that references **highlighted in bold green** can be found at the end of this document.

Response to Reviewer #1

Obomighie et al. performed an interesting piece of work aimed at enhancing our understanding of the mechanisms of pectin cross-linking and its impact on cell wall porosity using a combination of molecular dynamic simulations, mathematical modelling as well as experimental investigations. They provide an excellent background into the problem and chosen very advanced and sound techniques to address the problem ultimately providing strong evidence favouring a zipper model for pectin cross-linking. My only concern relates to the experimental biology side of the work where most of my expertise lies.

Our reply: We thank the Reviewer for their very positive assessment of our work.

Reviewer’s comment: *1. With regards to the aim of understanding the influence of HG cross-linking on porosity of the cell wall (page 9) and also given the novelty of the observations and claims herein, it may help to actually run simulations or perform experiments in arabidopsis plants with altered levels of pectin methyl esterification (similar to the plants used in the study by Weraduwege et al., 2016 (more details below)). The authors have indicated that redundancy of methyl esterification genes could be a limiting factor, but I am sure it will be helpful to at least see what happens even with small changes in the methyl esterification status of the wall. Details of the reference: Plant Physiol. 2016 Jun;171(2):833-48.doi: 10.1104/pp.16.00173 ox*

Our reply: We thank the Reviewer for this comment. We note that in our original manuscript we did run MD simulations to assess the impact of different degrees of methyl esterification on the cross-linking tendency of HG chains. Firstly, we investigated via metadynamics simulations the potential cross-linking mechanism via two methylated carboxyl groups in HG pectin (see Fig. 2). Secondly, we systematically explored the impact of methyl esterification re: the kinetics of cross-linking, see Table 1. The MD simulations provided more flexibility in exploring this question than is available through experimental work with arabidopsis mutants alone.

In this respect, the Reviewer refers to the redundancy of the pectin methyl esterase (PME) genes, of which there are around 67 **[1]** in arabidopsis. Mutants in many of these genes have no discernible phenotype; those that do, largely exert their effects only in very specific cell types (usually stomatal guard cells **[2,3]**). When conducting the experiments presented in our submitted manuscript, we originally tested *pme41*, an arabidopsis *pme* mutant that is unusual in that it has been reported to exhibit significant reductions in PME enzyme activity **[4]**. Even in this mutant, however, we saw no resultant effect on porosity using the same fluorescence quenching method we used on *sfr8* (See Fig. R1, next page).

“Understanding Pectin Cross-linking in Plant Cells Walls”
COMMSBIO-24-1998-T

Figure R1. (A) A time series comparable to the one shown in Fig. 4 of our original manuscript, comparing the fluorescent quenching in *pme41* and wild type. These data show the results of two separate sets of experiments combined. Differences in quenching efficiency are not significant. (B) The histogram shows there is no difference in quenching immediately after quencher addition.

We concluded from these data that it would be difficult to generate measurable perturbations in cell wall porosity through the use of *pme* mutants.

The Reviewer also suggests investigating the *cgr2 cgr3* double mutant described in [5], which is affected in HG pectin methylation through a direct effect on pectin methyl transferases (PMTs). This represented an attractive alternative approach to the use of the *pme* mutants and the problems of redundancy. This double mutant would be predicted to show reduced cell wall porosity as the less-methylated HG pectin should be more amenable to Ca^{2+} cross-linking. In response to the Reviewer’s suggestion, we have investigated this opportunity thoroughly. Because the mutant has severe growth defects, unlike *sfr8* (see photographs below in Fig. R2), the leaves of the mutant cannot be used to generate epidermal peels of the type we used in the original manuscript.

Figure R2. Five-week-old plants of Col0 wild type, and the *sfr8* and *cgr2 cgr3* mutants.

Thus, we investigated porosity in seedling roots, as originally described in the work of Liu *et al.* [6]. To capture fluorescence throughout an intact root, we used Lattice Lightsheet microscopy - due to its ability to rapidly image large volumes whilst maintaining resolution. This is preferable to confocal microscopy, where the same volume would take significantly longer with limited penetration thus reducing the number of roots analysed and membranes observed. We created maximum intensity projections of the fluorescence from root epidermal cells and found the quenching to be similar to that observed in wild type (Col 0) (see Fig. R3, next page).

“Understanding Pectin Cross-linking in Plant Cells Walls”
COMMSBIO-24-1998-T

Figure R3. *cgr2 cgr3* double mutant shows no difference in measured porosity using the fluorescence quenching method. (A) Maximum projections of three roots before (Fo) and after (F) the application of the fluorescence quencher. (B) Combined quenching efficiency data from multiple epidermal cells in each root.

In short, no difference in measurable porosity is observed in the PMT mutant. These data add to our evidence that mutants in pectin methyl esterification may not be appropriate for this study. We note that the fluorescence quenching method is one of a number of approaches that measure porosity on the basis of how large molecules (in this instance, the quencher) can move through the cell wall matrix. In some instances, factors other than cell wall pore size may contribute to the ability of molecules to travel. In the case of the *cgr2 cgr3* double knockout, published work [7] shows that in addition to a change in the proportion of methylated HG pectin, the sugar composition of pectin is altered, as is the *amount* of pectin. We predict that this is likely to have effects beyond the simple altering of crosslinking capacity. In particular, major alterations to HG pectin such as these, are likely to perturb the middle lamella, the HG-rich pectin boundary between adjacent cells. Evidence for this has been seen with other pectin methyltransferase mutants, such as *gosamt1 gosamt2*, which show a lack of cell adhesion as a result of this [8].

We suspected that the failure of the *cgr2 cgr3* double mutation to restrict the movement quencher molecules might be due to a confounding effect of the lack of normal adhesion between cells. We tested *cgr2 cgr3* using classic cell adhesion assays and discovered that like *gosamt1 gosamt2*, it too has a cell adhesion defect (as shown by the intense staining of the hypocotyl with ruthenium red, see Fig. R4, next page). We would predict that alterations to tissue permeability in the *cgr2 cgr3* mutant might outweigh any decrease in the porosity of the primary cell wall caused by the *cgr* mutations, rendering these mutants unhelpful for probing the relationship between cross-linking and porosity.

“Understanding Pectin Cross-linking in Plant Cells Walls”
COMMSBIO-24-1998-T

Figure R4. *cgr2 cgr3* shows a previously unreported cell adhesion defect. Seedlings are compared with Col0 (wild type) and *gosamt1 gosamt2* (a known cell adhesion mutant). (A) Both mutants show ruthenium red staining in the hypocotyl, indicative of a loss of cell adhesion. (B) Maximum projection of confocal z-stacks from 4-day old, etiolated seedling hypocotyls stained with propidium iodide. Red arrows reveal gaps between individual cells due to loss of cell adhesion.

These observations all add to the reasons we opted to use the *sfr8* mutant to measure and model the effect pectin crosslinking has on cell wall porosity. Unlike the HG methylation mutants, *sfr8* (like its allelic *mur1* mutants) has a mutation that leads to defective RGII pectin crosslinking, experimentally proven in earlier studies [9,10]. In addition, it is long-established in the literature that reduced RGII pectin crosslinking leads to increased porosity in other plant species [11]. This provided a firm basis for studying the link between changes in pectin crosslinking and cell wall porosity. Use of *sfr8* also avoids other confounding factors; the mutant shows no obvious growth defect (see Fig. R2 above) so is more readily compared with wild type and does not cotyledon fusion or other abnormal cell rearrangements or protuberances indicative of cell dissociation or sloughing off [12].

Therefore, in light of this experimental evidence, we can confidently say that MD simulations represent the most effective option to explore the relationship between pectin modification and crosslinking - whilst the use of the *sfr8* mutant is best suited to the experimental investigation of how crosslinking relates to porosity. We have added depth to our explanation of the issues and the reasons for the choice of *sfr8* in the revised version of the manuscript.

Changes made: We have added the following discussion in the revised version of the manuscript, on page 9, lines 259-278: “We remark at this stage that a large number of genes control pectin methyl esterification (and thus, have the potential to affect pectin cross-linking) and redundancy in their function means that genetic mutants are largely unaffected by loss of function of any single PME gene [16,17]. In contrast, mutants that are affected globally in HG pectin methyl esterification exhibit extreme structural defects, including reduced growth and loss of cell adhesion, due to their effects on the HG-rich middle lamella [27]. This makes it difficult to draw conclusions about cross-linking by comparing them with wild type plants.

To circumvent this issue whilst making appreciable alterations to pectin cross-linking, we have used a well-studied genetic mutant of *Arabidopsis thaliana* which exhibits qualitative differences in its RGII pectic domain but shows unaltered amounts of pectin and no defect in cell adhesion [34]. This mutation decreases the formation of boron bridges between pectin chains. RG-II is covalently linked to HG pectin [19] and in wild type plants, those boron bridges draw pectin chains closer together and facilitate further cross-linking via the mechanisms we have examined in our simulations [30, 35]. As the formation of boron bridges and their knock-on effect on the creation of other linkages between pectin

“Understanding Pectin Cross-linking in Plant Cells Walls”
COMMSBIO-24-1998-T

chains, (e.g., HG-Ca²⁺ bridges) are intertwined and cannot be disentangled experimentally, we focus on understanding their collective impact on the porosity of the cell wall.”

Reviewer’s comment: 2. Perhaps not a very big deal, but could the authors explain why the quenching level for the lower *sfr8* mutant was similar to *Col(0)*, i.e. will this more frequently if they tested more samples and if so what could possibly be the reason.

Our reply: The Reviewer is correct to point out that the levels of quenching between samples vary. This is why we reported several images for both *Col0* and *sfr8* in Fig.4, panel a), of our original manuscript. However, quenching is consistently more efficient in the *sfr8* mutant than in the wild type. We remark that we have quantified the fluorescence by considering numerous samples: 39 for *Col0* and 42 for *sfr8* (we only show three sets of samples for either *Col0* or *sfr8* in Fig.4, panel a), of our original manuscript.

Changes made: To avoid confusion, we have amended Fig.4 by replacing the lower image with a different sample (see Fig. R5). The graph still represents the level of variation in the response that we see between different samples.

Figure R5. Experimental measurements of the cell wall porosity in *Col-0* wild type and *sfr8* arabidopsis leaves. a) Change in the fluorescence of the FM4-64-stained plasma membrane (see text) for *Col-0* and *sfr8* cell wall samples. A time series of two representative fluorescence images for each genotype are shown. The optical microscopy images were taken before quenching (F_0), immediately after addition of trypan blue (F) and at three-minute intervals thereafter. Three representative images are reported for each point in time. The scale bar (yellow) is 5 μ m. b) Change in the relative fluorescence (F/F_0) as a function of time. The mean values of F/F_0 are measured every three minutes within a 0–15-minute time interval, for both *Col-0* and *sfr8*. The value at the Y-axis intersection represents the unquenched sample (F/F_0) and is equivalent to 1.0. The “*” and “**” symbols indicate moderate (P value 0.05) and strong (P value 0.01) evidence, respectively for a significant statistical difference between the two-time series. c) SEM images of *Col-0* and *sfr8* cell walls. Three representative images, taken from 3 different leaves, are shown for both genotypes. The scale bar (yellow) is 0.25 μ m. d) Pore size (area, nm²) frequency distributions for *Col-0* and *sfr8*.

Reviewer’s comment: 3. For future reviews it may also be helpful for authors to number text lines to help with referencing.

Our reply: We appreciate that line numbering will facilitate the reviewing process.

Changes made: We have added line numbering in the revised version of the main text and of the supplementary information (SI).

Response to Reviewer #2

This study utilized a multidisciplinary approach including in silico approaches and experimental testing to provide novel insights into the mechanism of cross-linking of pectin and revealed a strong link between HG cross-linking and porosity of the cell wall. The work is intriguing, especially in the context of our new understanding of the plant primary cell wall model, and the supporting evidence are illustrative and engaging. Overall, the research quality is good, there is sufficient data presented in the results and discussed afterwards. However, additional editing is needed to enhance quality. The following are the three most important improvements that the author needs to make.

Our reply: We thank the Reviewer for their very positive assessment of our work.

Reviewer’s comment: 1. The arabidopsis cell wall mutant sensitive-to-freezing-8 (*sfr8*) in which RGII pectin cross-linking is reduced was selected for experimental validation. The work indicated that disrupted RGII boron bridges affected HG-Ca²⁺ crosslinking, which further influenced pectin properties including cell wall porosity. HG is the most abundant pectin subtype in plant cell wall, while RGII only makes up a small percentage of pectin. I was wondering why not use cell wall mutants where HG cross-linking is directly disrupted for experimental validation? Can this point be explained in greater detail?

Our reply: We fully appreciate the Reviewer’s point that it would seem obvious to attempt to perturb HG crosslinking directly. Our detailed response to Reviewer’s comment: 1. of Reviewer #1 and the additional data we have now gathered provide the answer the question of why we did not do this - and why we found an RG-II crosslinking mutant to be more suitable for our study.

The Reviewer makes another important point, that RG-II makes up only a small proportion of total pectin. It is possible that affecting a minority component of pectin in this way is less prone to the compensatory mechanisms that occur when we attempt to disrupt HG, the major component of pectin. In addition, RG-II crosslinking is partially, but not entirely, abolished in *mur1/sfr8* mutants [10,9]. This means it is possible to examine experimentally the effects of altered crosslinking without making very severe changes to the cell wall. Whilst it is true that RG-II is a minority component of pectin, its crosslinking status makes a huge difference to the properties of the cell wall, as has been reported in many papers (reviewed in Ref. [9]). RG-II is covalently linked to HG [13] and its borate-crosslinking status impacts upon the whole pectin network, allowing this change to make a significant impact. We have incorporated this discussion in the revised version of the manuscript.

“Understanding Pectin Cross-linking in Plant Cells Walls”
COMMSBIO-24-1998-T

Changes made: We have added the following discussion in the revised version of the manuscript, on page 9, lines 259-278: “We remark at this stage that a large number of genes control pectin methyl esterification (and thus, have the potential to affect pectin cross-linking) and redundancy in their function means that genetic mutants are largely unaffected by loss of function of any single PME gene [16,17]. In contrast, mutants that are affected globally in HG pectin methyl esterification exhibit extreme structural defects, including reduced growth and loss of cell adhesion, due to their effects on the HG-rich middle lamella [27]. This makes it difficult to draw conclusions about cross-linking by comparing them with wild type plants.

To circumvent this issue whilst making appreciable alterations to pectin cross-linking, we have used a well-studied genetic mutant of *Arabidopsis thaliana* which exhibits qualitative differences in its RGII pectic domain but shows unaltered amounts of pectin and no defect in cell adhesion [34]. This mutation decreases the formation of boron bridges between pectin chains. RG-II is covalently linked to HG pectin [19] and in wild type plants, those boron bridges draw pectin chains closer together and facilitate further cross-linking via the mechanisms we have examined in our simulations [30, 35]. As the formation of boron bridges and their knock-on effect on the creation of other linkages between pectin chains, (e.g., HG-Ca²⁺ bridges) are intertwined and cannot be disentangled experimentally, we focus on understanding their collective impact on the porosity of the cell wall.”

Reviewer’s comment: 2. *In the section of “Conclusions”, the authors mentioned that predicting the interactions between pectin chains in different contexts and forecasting changes in porosity has many applications in agrotechnology, followed by giving several examples from previous studies. However, these references cited by the authors seem too general. I suggest citing several more updated references which can specifically reveal a strong relationship between cell wall porosity and stress tolerance etc.*

Our reply: We agree that there was much room for improvement here. We have followed the Reviewer’s advice and updated this section with more specific references.

Changes made: We have added the following discussion in the revised version of the manuscript, on pages 17 and 18, lines 471-485: “The ability to predict the interactions between pectin chains in different contexts and forecast changes in porosity has many applications in agritechology. Pectin structure determines the level of success attacking fungal and bacterial pathogens may have [8, 50, 51] and mechanisms have even been identified whereby the invading fungus targets Ca-pectate domains to render the wall more porous and facilitate access to the cell’s nutrients [52]. Growing evidence suggests that remodelling of cell wall and particularly pectin structure is involved in the defence against abiotic stresses [53-55] and there are reports of wall porosity affecting susceptibility to ion toxicity and dehydration [51, 56, 57]. Altering the porosity of cell wall has also been shown to increase the efficiency of CO₂ uptake, increasing photosynthesis as a result [58]. In addition, differences in cell wall porosity can be exploited in optimization of nanoparticle uptake [59], wood quality and strength [56] and saccharification of plant and algal material to produce biofuels [10, 61, 62]. There are clearly opportunities for crop protection here.”

Reviewer’s comment: 3. *The authors made a conclusion that this study challenge the prevailing egg-box model, favoring a zipper model for pectin cross-linking instead. The conclusion was made only based on molecular simulations. I was wondering if any experimental data can be provided to support this conclusion*

Our reply: Small-angle X-ray scattering (SAXS) has been previously employed to investigate the structural details of Ca²⁺ crosslinking in pectins and alginates (see Refs. [14,15]). Although we do not

have access to this particular experimental technique, it is important to note that SAXS does not directly support the egg-box model. Notably, even recent experimental work on arabidopsis (see Ref. [16]) cites previous modeling efforts, specifically Ref. [17], as evidence for the egg-box model. Interestingly, Ref. [17] does not advocate the original egg-box model but suggests a “shifted egg-box” configuration for galacturonate and guluronate chains. In fact, growing computational evidence challenges the conventional egg-box model (see Refs. [18,19]). Thus, while we cannot provide experimental data to support our findings, our computational results contribute to the increasing body of evidence questioning the validity of the egg-box model for pectin.

Changes made: We have clarified this aspect in the revised version of the manuscript, by adding the following discussion on page 5, lines 125-132: “Small-angle X-ray scattering (SAXS) has been previously employed to investigate the structural details of Ca²⁺ crosslinking in pectins and alginates [28, 29]. However, it is important to note that SAXS cannot provide direct evidence for the egg-box model. Notably, even recent experimental work on arabidopsis [30] cites previous modeling efforts, specifically Ref. 31, as evidence for the egg-box model. Interestingly, Ref. 31 does not advocate the original egg-box model but suggests a “shifted egg-box” configuration for galacturonate and guluronate chains.”

Response to Reviewer #3

The manuscript by Obomighie et al. used a multidisciplinary approach spacing from MD simulations to experimental assays to clarify the cross-linking of Pectin and the relationship between its structure and the porosity of the cell wall. This paper is well written, and the conclusions are supported by their analyses, thus representing a step forward in understanding the molecular basis of such a mechanism. I can recommend the publication of the manuscript after the following points will be addressed:

Our reply: We thank the Reviewer for their very positive assessment of our work.

Reviewer’s comment: *1 - Can the authors provide a brief description in the SI of the initial simulations they run using the TIP4P/Ice model and a quantitative explanation of why they switched to the TIP4P one?*

Our reply: Initially, we have performed molecular dynamics (MD) simulations of HG in water utilising the CHARMM36 force field (HG chains) in combination with the TIP4P/Ice water model (water molecules). This is because previous work by us as well as other authors [20–22] has demonstrated the validity of this particular combination of force fields in describing the interactions of bio molecules with water, especially when working with supercooled liquid water (which we intend to explore in the context of HG in future work).

However, at ambient temperature and pressure, the (self-)diffusion coefficient of TIP4P/Ice water is 1.2 nm²/ns [23], which is substantially lower than the experimental value of 2.3 nm²/ns [24]. In contrast, the diffusion coefficient of (the original) TIP4P water model at the same conditions of temperature and pressure is 2.4 nm²/ns [25], which is much closer to the experimental value.

Whilst we found that the slower water dynamics of the TIP4P/Ice water model compared to that of the TIP4P water model does not have any impact on the results reported in this work, using the TIP4P/Ice water model is much more computationally expensive than using the TIP4P water model, particularly when studying the aggregation of long HG chains in water. This is because the slow diffusion of

TIP4P/Ice water results in a slower diffusion of the HG chains as well, which in turns leads to much longer simulations times to observe the aggregation process.

For the purposes of offering a quantitative comparison, we report in Fig. R6 (which we have also added to the revised version of the SI) the mean square displacement (MSD) of (the centre of mass of) the HG chains, computed for the same HG/water system (8 HG chains, 40-unit each – i.e., the same system depicted in Fig. 3 in the main text) at ambient temperature and pressure, as a function of time - using either the TIP4P or the TIP4P/Ice water model. The resulting diffusion coefficients, 0.19 and 0.05 nm²/ns for HG in TIP4P and TIP4P/Ice water, respectively, demonstrate the impact of the slow dynamics of the TIP4P/Ice water at room temperature.

Figure R6. Mobility of HG chains in different water models. Mean squared displacement for the HG chains in either TIP4P or TIP4P/Ice water. These results have been computed for systems containing 8 40-unit HG chains solvated in ~200,000 water molecules, over 30 ns long molecular dynamics trajectories.

Changes made: We have added a whole new section (Section 1.2) in the SI, discussing this aspect in detail.

Reviewer’s comment: 2 - *The authors should provide more details about the metadynamics simulations: i) what's the length of such simulations? ii) they should provide a better explanation of why they finally used those specific values of width, height and deposition stride of the Gaussian potentials; iii) are they sure that the convergence was reached?*

Our reply: We thank the Reviewer for bringing to our attention the need to clarify several details re: the metadynamics simulations we have performed. Specifically:

- I. The length of the metadynamics simulations reported in Fig. 2 ranged from 500 to 700 ns. Whilst the concept of time in metadynamics simulations does not have a direct physical meaning (as the time evolution of the system is heavily influenced by the artificial bias introduced for the purposes of exploring the free energy surface of interest), such long

simulation times were necessary to ensure the proper convergence of the resulting free energy surface - as pointed out by the Reviewer.

- II. In order to ensure that we have indeed reached convergence, we have: (i.) monitored the changes of the free energy surface as a function of the simulation time; (ii.) explored the impact of varying the height and width of the Gaussian hills (i.e., the bias potential) – we have found that within reasonable ranges (motivated below) the changes in the resulting free energy surfaces lie within the uncertainty we have quantified via the reweighing technique of Tiwary and Parrinello [26] (see next point); (iii.) utilised the reweighing technique of Tiwary and Parrinello [26] to provide a quantitative estimate of the error relative to the convergence of the free energy surface. As such, we are confident we have deployed a robust portfolio of control measures to ensure the convergence of the free energy surfaces reported in Fig. 2.
- III. We have chosen the height (W), width (σ), deposition rate (n_s) and bias factor (γ) re: to the Gaussian hills (i.e., the bias potential) as follows:
- The height (W) was initially set to half the value of the thermal energy ($k_B T$) at room temperature, i.e., 1.24 kJ/mol. This is a common rule of thumb in terms of striking a compromise between sampling efficiency and accuracy. We have explored values of W between 1.00 and 2.00 kJ/mol, in conjunction with bias factors (γ , which value also influences said efficiency/accuracy ratio) ranging from 50 to 100. We have eventually settled on $W=1.4$ and 1.2 kJ/mol for investigating the calcium bridge (Fig. 2c) and the hydrogen bond interactions (Fig. 2e and Fig. 2g), respectively, in conjunction with $\gamma=60$. This choice enabled us to converge our free energy surfaces (within the uncertainty computed via the reweighing technique of Tiwary and Parrinello [26] and reported in Fig.2 as the shaded blue regions) within an acceptable simulation time (500-700 ns). Any combinations of the values of W and γ within the above-mentioned ranges led to free energy surfaces within said uncertainty.
 - The width (σ) was initially set to 0.01 nm. This value turned out to be too small, in that the resulting free energy surfaces struggled to achieve convergence due to the too-fine resolution imposed by such a small width. Thus, we have increased σ to 0.04 and 0.05 nm for investigating the calcium bridge (Fig. 2c) and the hydrogen bond interactions (Fig. 2e and Fig. 2g), respectively. Values of $\sigma > 0.08$ nm resulted in free energy surfaces lacking sufficient detail/resolution.
 - The deposition rate was set to 500 MD steps (equivalent to 1 ps), as the structural relaxation time of TIP4P water at room temperature and pressure is of the order of 10^{-1} ps [27]. We have found this deposition rate to lead to the efficient sampling of the free energy surface, specifically in comparison to 1000 MD steps, a deposition rate we have trialled as well (but abandoned in the interest of computational efficiency).
 - We discussed the choice of bias factor (γ) above, in conjunction with the choice of the height (W).

Changes made: The changes we made to the manuscript are (on page 21, lines 612-639): “The deposition rate of the bias was set to 500 MD steps (equivalent to 1 ps) as the structural relaxation time of TIP4P water at room temperature and pressure is of the order of 10^{-1} ps [81].

The width of the Gaussian potentials was initially set to $\sigma = 0.01$ nm. This value turned out to be too small, in that the resulting free energy surfaces struggled to achieve convergence due to the too-fine resolution. Thus, we have increased σ to 0.04 and 0.05 nm for investigating the calcium bridge and the hydrogen bond interactions, respectively. Values of $\sigma > 0.08$ nm resulted in free energy surfaces lacking sufficient detail/resolution.

W was initially set to half the value of the thermal energy ($k_B T$) at room temperature, i.e., 1.24 kJ/mol. This is a common rule of thumb in terms of striking a compromise between sampling efficiency and accuracy. We have explored values of W between 1.00 and 2.00 kJ/mol, in conjunction with bias factors (which value also influences said efficiency/accuracy ratio) ranging from 50 to 100. Any combination of the values of W and γ within the above-mentioned ranges led us to obtain free energy surfaces within the uncertainty computed via the reweighing technique of Tiwary and Parrinello [82] (and reported in Fig. 2 as the shaded blue regions) within an acceptable simulation time (500-700 ns). Whilst the concept of time in metadynamics simulations does not have a direct physical meaning (as the time evolution of the system is heavily influenced by the artificial bias introduced for the purposes of exploring the free energy surface of interest), such long simulation times were necessary to ensure the proper convergence of the resulting free energy surface. To this end, we have: (i.) monitored the changes of the free energy surface as a function of the simulation time; (ii.) explored the impact of varying the height and width of the bias potential; (iii.) utilised the reweighing technique of Tiwary and Parrinello [82] to provide a quantitative estimate of the uncertainty associated with the free energy surface.”

References (Rebuttal)

- [1] J. Harholt, A. Suttangkakul, H. Vibe Scheller, *Plant Physiol.* 153 (2010) 384–395.
- [2] S. Amsbury, L. Hunt, N. Elhaddad, A. Baillie, M. Lundgren, Y. Verhertbruggen, H.V. Scheller, J.P. Knox, A.J. Fleming, J.E. Gray, *Curr Biol* 26 (2016) 2899–2906.
- [3] Y.C. Huang, H.C. Wu, Y.D. Wang, C.H. Liu, C.C. Lin, D.L. Luo, T.L. Jinn, *Plant Physiol* 174 (2017) 748–763.
- [4] T. Qu, R. Liu, W. Wang, L. An, T. Chen, G. Liu, Z. Zhao, *Cryobiology* 63 (2011) 111–117.
- [5] S.M. Weraduwege, S.-J. Kim, L. Renna, F.C. Anozie, T.D. Sharkey, F. Brandizzi, *Plant Physiol.* 171 (2016) 833.
- [6] X. Liu, J. Li, H. Zhao, B. Liu, T. Günther-Pomorski, S. Chen, J. Liesche, *J. Cell Biol.* 218 (2019) 1408–1421.
- [7] S.-J. Kim, M.A. Held, S. Zemelis, C. Wilkerson, F. Brandizzi, *Plant J.* 82 (2015) 208–220.
- [8] H. Temple, P. Phyo, W. Yang, J.J. Lyczakowski, A. Echevarría-Poza, I. Yakunin, J.P. Parra-Rojas, O.M. Terrett, S. Saez-Aguayo, R. Dupree, A. Orellana, M. Hong, P. Dupree, *Nat. Plants* 8 (2022) 656–669.
- [9] M.A. O’Neill, T. Ishii, P. Albersheim, A.G. Darvill, *Annu. Rev. Plant Biol.* 55 (2004) 109–139.
- [10] P.E. Panter, O. Kent, M. Dale, S.J. Smith, M. Skipsey, G. Thorlby, I. Cummins, N. Ramsay, R.A. Begum, D. Sanhueza, S.C. Fry, M.R. Knight, H. Knight, *New Phytol* 224 (2019) 1518–1531.
- [11] A. Fleischer, M.A. O’Neill, R. Ehwald, *Plant Physiol* 121 (1999) 829–838.
- [12] P.E. Panter, J. Seifert, M. Dale, A.J. Pridgeon, R. Hulme, N. Ramsay, S. Contera, H. Knight, *J. Exp. Bot.* 74 (2023) 2680.
- [13] T. Ishii, T. Matsunaga, *Phytochemistry* 57 (2001) 969–974.
- [14] K.I. Draget, B.T. Stokke, Y. Yuguchi, H. Urakawa, K. Kajiwara, *Biomacromolecules* 4 (2003) 1661–1668.
- [15] I. Ventura, J. Jammal, H. Bianco-Peled, *Carbohydr. Polym.* 97 (2013) 650–658.
- [16] D. Shi, J. Wang, R. Hu, G. Zhou, M.A. O’Neill, Y. Kong, *Plant Mol. Biol.* 94 (2017) 267–280.
- [17] I. Braccini, S. Pérez, *Biomacromolecules* 2 (2001) 1089–1096.
- [18] W. Plazinski, *J. Comput. Chem.* 32 (2011) 2988–2995.

“Understanding Pectin Cross-linking in Plant Cells Walls”
COMMSBIO-24-1998-T

- [19] A. Assifaoui, A. Lerbret, H.T.D. Uyen, F. Neiers, O. Chambin, C. Loupiac, F. Cousin, *Soft Matter* 11 (2014) 551–560.
- [20] H. Lee, *PLOS ONE* 13 (2018) e0198887.
- [21] K. Mochizuki, V. Molinero, *J. Am. Chem. Soc.* (2018).
- [22] G.C. Sosso, T.F. Whale, M.A. Holden, P. Pedevilla, B.J. Murray, A. Michaelides, *Chem. Sci.* 9 (2018) 8077–8088.
- [23] Ł. Baran, W. Rżysko, L.G. MacDowell, *J. Chem. Phys.* 158 (2023) 064503.
- [24] K. Krynicky, C. D. Green, D. W. Sawyer, *Faraday Discuss. Chem. Soc.* 66 (1978) 199–208.
- [25] D.V. Zlenko, *Biophysics* 57 (2012) 127–132.
- [26] P. Tiwary, M. Parrinello, *J. Phys. Chem. B* 119 (2015) 736–742.
- [27] P. Gallo, M. Rovere, *J. Chem. Phys.* 137 (2012) 164503.

Response to Editor and Reviewers' feedback

Since Reviewer #2 could not provide any further feedback, we asked Reviewer #1 to have a look at your rebuttal, addressing Reviewer #2's comments. They suggested that you make clear in the abstract that while a variety of methods were employed, the model is most strongly supported by the molecular dynamics simulations. We highly recommend that these edits are made.

We have amended the abstract of the revised version of the manuscript as suggested. Specifically, the abstract now reads: “Pectin is a major component of plant cells walls. The extent to which pectin chains crosslink with one another determines crucial properties including cell wall strength, porosity, and the ability of small, biologically significant molecules to access the cell. Despite its importance, significant gaps remain in our comprehension, at the molecular level, of how pectin cross-links influence the mechanical and physical properties of cell walls. This study employs a multidisciplinary approach, combining molecular dynamics simulations, experimental investigations, and mathematical modelling, to elucidate the mechanism of pectin cross-linking and its effect on cell wall porosity. The computational aspects of this work challenge the prevailing egg-box model, favouring instead a zipper model for pectin cross-linking, whilst our experimental work highlights the significant impact of pectin cross-linking on cell wall porosity. This work advances our fundamental understanding of the biochemistry underpinning the structure and function of the plant cell wall. This knowledge has important implications for agricultural biotechnology, informing us about the chemical properties of plant pectins that are best suited for improving crop resilience and amenability to biofuel extraction by modifying the cell wall”.

Please explicitly and clearly state whether and what Statistical tests (if any, for example for Fig 4b) were done in the Statistics and Reproducibility statement. At the moment this section of your methods refers the reader to a brief discussion of reproducibility in the Supplementary Information.

With respect to the fluorescence measurements reported in Fig.4b, three separate regions of interest (ROI) were assessed per epidermal peel (for a total of 39 and 42 ROI for Col0 = 39 and *sfr8*, respectively). The data was combined from 3 separate experiments. For statistical comparisons, a two-way ANOVA with Bonferroni's multiple comparisons test¹ was employed. Significance was set at $\alpha = 0.05$.

With respect to the SEM measurements, the pore size distributions from different ROI (9 and 11 ROI for Col0 and *sfr8*, respectively) of 3 individual plants per genotype was assessed. A frequency distribution with a bin width and range of 15 and 5-305 nm², respectively, was employed. The analysis was performed using GraphPad Prism version 10.1.0. Whilst the leaf to leaf variation in pore sizes varied noticeably (hence it was not appropriate to compare mean pore sizes directly across the two genotypes), we gauge ϕ to be of the order of ≈ 0.05 for both samples, with $\phi_{\text{wild type}} < \phi_{\text{sfr8}}$.

In terms of the computational aspects of the work, we have expanded on the discussion on the reproducibility of our molecular dynamics simulations (in the Supplementary Information) by

“Understanding Pectin Cross-linking in Plant Cell Walls”
COMMSBIO-24-1998A

providing the details of the reweighting technique we have utilised to assess the uncertainty related to the results of the metadynamics simulations.

We have added all these details to the “Statistics and Reproducibility” section of the revised manuscript.

Please submit an updated MDS checklist whereby you add a table like the one described in 4a of the checklist with the necessary information and also specify where the initial coordinate and simulation input files and a coordinate file of the final output (point 4d) are provided.

We have updated the MDS checklist. Specifically, we have now added into the Supplementary Information two tables: Table 1, which refers to the molecular dynamics simulations of HG aggregation, and Table 3, which refers to the metadynamics simulations of HG crosslinking. In these tables we have provided all the requested computational details. We have also elaborated on the discussion on Table 2, which serves to ensure the robustness and reproducibility of our simulations.

In addition, we have gathered all the input, coordinates and output files re: every molecular dynamics and metadynamics simulation we have performed. This data can now be found in a publicly available GitHub repository (PEC_LINK, https://github.com/gcsosso/PEC_LINK.git). We have also linked a release of said repository to a Zenodo repository (also publicly accessible), for which we have obtained a permanent DOI: <https://zenodo.org/records/14366069>.

The link <https://wrap.warwick.ac.uk/id/eprint/188255/> currently indicates "page not found". Please provide the source data (e.g. simulation input and output files) as supplementary data files or deposit them in a suitable repository with a corresponding DOI (as suggested in the attached table). You can include the information about where these files can be found in the Data Availability Statement.

We have gathered all the source data, including all the input and output files, as well as the data we have used to plot each of the figure in the main paper. This data can now be found in a publicly available GitHub repository (PEC_LINK, https://github.com/gcsosso/PEC_LINK.git). We have also linked a release of said repository to a Zenodo repository (also publicly accessible), for which we have obtained a permanent DOI: <https://zenodo.org/records/14366069>.

Finally, please ensure you provide the numerical data for the graphs in Fig. 4b and d.

We have provided the numerical data for the graphs in Fig. 4b and Fig. 4d. Said data can be found in the same Zenodo repository mentioned above (DOI: <https://zenodo.org/records/14366069>)

“Understanding Pectin Cross-linking in Plant Cell Walls”
COMMSBIO-24-1998A

- (1) McHugh, M. L. Multiple Comparison Analysis Testing in ANOVA. *Biochem. Medica* **2011**, 21 (3), 203–209. <https://doi.org/10.11613/BM.2011.029>.